# TDBench: Benchmarking Vision Language Models on Top-Down Image Understanding

## Abstract

Top-down images play an important role in safety-critical settings such as autonomous navigation and aerial surveillance, where they provide holistic spatial information that front-view images cannot capture. Despite this, Vision Language Models (VLMs) are mostly trained and evaluated on front-view benchmarks, leaving their performance in the top-down setting poorly understood. Existing evaluations also overlook a unique property of top-down images: their physical meaning is preserved under rotation. In addition, conventional accuracy metrics can be misleading, since they are often inflated by hallucinations or "lucky guesses", which obscures a model's true reliability and its grounding in visual evidence. To address these issues, we introduce TDBench, a benchmark for top-down image understanding that includes 2000 curated questions for each rotation. We further propose RotationalEval (RE), which measures whether models provide consistent answers across four rotated views of the same scene, and we develop a reliability framework that separates genuine knowledge from chance. Finally, we conduct four case studies targeting underexplored real-world challenges. By combining rigorous evaluation with reliability metrics, TDBench not only benchmarks VLMs in top-down perception but also provides a new perspective on trustworthiness, guiding the development of more robust and grounded AI systems.

## 1 Introduction

Top-down images provide comprehensive spatial overviews and clear geometric context, supporting tasks such as autonomous navigation, aerial surveillance, mapping, and disaster assessment (Lu et al., 2018; Nearmap, 2022; Zhao et al., 2025). Top-down images from drones or satellites provide a complete "bird's-eye" view, offering several unique advantages over conventional front-view images: they reduce occlusion between objects, maintain more consistent scale across the frame, and reveal complete spatial layouts that are impossible to observe from ground level. These properties allow analysts or autonomous systems to reason about large geographic areas efficiently, which is essential in applications such as traffic monitoring, urban planning, and environmental response.

Despite their importance, top-down images are substantially underrepresented in the datasets commonly used to train and evaluate Vision Language Models (VLMs). Well-known datasets such as COCO (Lin et al., 2015) and ImageNet (Russakovsky et al., 2015) contain primarily front-view images, where appearance cues, object sizes, and spatial relationships are largely different from aerial perspectives. For instance, in our preliminary data audit, fewer than 7% images (595 of 8,629) from the VisDrone dataset(Zhu et al., 2021) could be considered truly top-down. This limited coverage leaves current VLMs largely untested for top-down understanding, even though such models are increasingly applied in drone-based and remote-sensing systems.

Most existing VLM benchmarks (Liu et al., 2024b; Yue et al., 2024; Yu et al., 2024; Lu et al., 2024) are not designed for top-down images. While these benchmarks have driven progress in general-purpose visual reasoning, they provide little insight into how VLMs handle the distinct challenges of top-down perception. Aerial scenes present small, densely packed objects, drastically different viewing angles, and weak perspective depth cues. Contextual cues that aid object recognition in conventional images may be absent or transformed in top-down perspectives. VLMs trained mostly on canonical-view data often fail to generalize to these conditions, leading to severe accuracy drops (Danish et al., 2025;

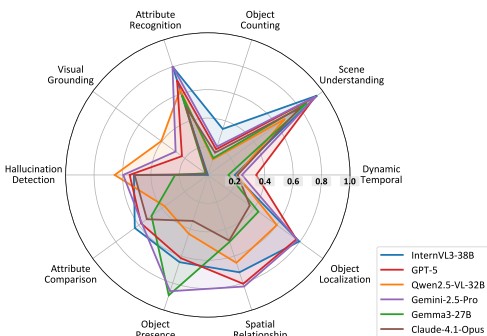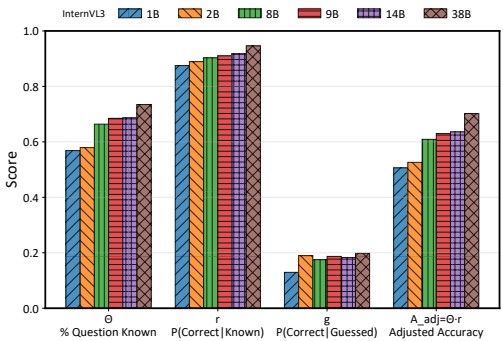

Figure 1: **(Left)** Accuracy across ten top-down image tasks in TDBench. **(Right)** Knowledge decomposition analysis from TDBench: *% of questions known ($\theta$)* measures the proportion of questions a model truly knows; $P(Correct|Known)$ *(r)* is the model's accuracy among the questions that it knows; $P(Correct|Guessed)$ *(g)* is the model's accuracy among the questions it does not know; and the *Adjusted Accuracy ($A_{adj} = \theta \cdot r$)* is the model's accuracy without lucky guesses.

Li et al., 2024a). Without a dedicated benchmark, it is difficult to measure or systematically improve their performance on top-down views.

To address this gap, we present TDBench, a benchmark for evaluating VLMs on top-down image understanding. TDBench contains 2,000 carefully constructed questions drawn from public aerial datasets and high-fidelity simulations, covering diverse settings and tasks relevant to real-world operations. We also introduce RotationalEval (RE), an evaluation method that leverages a key property of top-down images: their physical meaning is preserved under rotation. Unlike front-view images, where rotation produces implausible scenes (for example, the sky appearing below or objects upside down), rotating a top-down image is equivalent to changing a drone's heading, so the scene remains physically consistent. RE tests whether models can answer correctly across all four rotated views, recognizing that semantics and object identities remain the same while spatial descriptors (e.g., "top left"), and coordinates legitimately change. This provides a stricter and more diagnostic measure of visual reasoning, reducing the influence of spurious one-off successes.

Vision Language Models (VLMs) often hallucinate, generating answers from learned text patterns instead of grounding them in the provided image (Li et al., 2023b; Bai et al., 2025b). This can artificially inflate scores under conventional evaluation. However, an ungrounded guess is highly unlikely to be correct across four different rotations, RE naturally filters out these successes. We further formalize this with new reliability-oriented metrics that disentangle a model's visually-grounded knowledge from its apparent accuracy. This provides a more quantitative view of model trustworthiness than raw accuracy alone.

Finally, we conduct four application-oriented case studies for real-world applications: digital and physical "zoom-in", handling partially visible objects and reasoning about depth from 2D views. These case studies demonstrate how TDBench can guide the design and deployment of VLM-based aerial systems. In summary, our main contributions are:

- **Application-driven Benchmark.** We build TDBench, a top-down benchmark of **2,000** question–answer pairs from public datasets and high-fidelity simulation, organized into ten evaluation dimensions. To demonstrate its practical relevance, we also conduct **four case studies** that examine VLMs on real-world aerial applications, providing actionable insights for deployment.

- **Rotation-invariant Evaluation.** We introduce **RotationalEval (RE)**, an evaluation strategy that requires consistent answers across four rotated views of each image. By requiring models to be rotationally consistent, correctly adapting their spatial reasoning to each orientation, RE provides a far more robust and diagnostic measure of their performance than single-view evaluation.

- **Probability-based Knowledge Reliability Analysis.** Beyond raw and RE accuracy, we propose a **probabilistic analysis** that decomposes model performance into *% of questions known ($\theta$)*, $P(Correct|Known)$ *(r)*, $P(Correct|Guessed)$ *(g)*, and further aggregate them into *Adjusted Accuracy ($\theta \cdot r$)*, which reveals how much of a model's apparent correctness stems from genuine knowledge rather than lucky guesses.

## 2 RELATED WORKS

### 2.1 VISION LANGUAGE MODELS (VLMS)

Vision Language Models (VLMs) extend large language models (LLMs) to visual inputs by aligning image features with text representations. Most current VLMs adopt a two-stage design: a pretrained visual encoder (e.g., CLIP (Radford et al., 2021) or SigLIP (Zhai et al., 2023)) is coupled with a pretrained text-only LLM via a learnable projection module, as in LLaVA (Li et al., 2024b) and InternVL (Chen et al., 2025). This setup preserves the language backbone while enabling it to interpret visual features. Some models instead use early-fusion architectures that train perception and language components jointly, strengthening visual grounding and cross-modal reasoning. Proprietary models such as GPT (OpenAI, 2024), Gemini (Google, 2024), and Claude (Anthropic, 2024) may follow similar multimodal principles at larger scales.

VLMs are generally trained on large-scale image–text pairs from datasets like LAION (Schuhmann et al., 2022), COCO (Lin et al., 2015), and ImageNet (Russakovsky et al., 2015), which may contain few top-down images and thus treat them as out-of-distribution (OOD). While this broad training enables rich visual–linguistic knowledge, it biases models toward ground-level scenes and object appearances. As a result, their generalization to top-down views, where objects appear smaller, depth cues are weak, and spatial relationships dominate, remains underexplored, motivating the need for a dedicated benchmark.

### 2.2 VLM BENCHMARKS

Recent years have seen the emergence of numerous benchmarks for evaluating Vision–Language Models (VLMs) on diverse multimodal reasoning tasks. General-purpose benchmarks such as MMBench (Liu et al., 2024b), MMMU (Yue et al., 2024), MME (Fu et al., 2024), and MM-Vet (Yu et al., 2024) assess general knowledge, visual perception, commonsense reasoning, and spatial understanding. However, these benchmarks focus primarily on conventional front-view imagery and include few tasks involving aerial or top-down perspectives. They thus overlook challenges unique to top-down understanding, including extreme scale variation, weak depth cues, and dense spatial layouts, which often cause VLMs to underperform on aerial tasks.

A few recent efforts have begun addressing this gap using remote sensing images. For example, Hu et al. (2023), Muhtar et al. (2024), Kuckreja et al. (2023), and Danish et al. (2025) evaluate VLMs on satellite data. These datasets mostly comprise low-resolution images (meters per pixel) aimed at large-scale land cover classification or scene categorization. They rarely involve human-scale and near-surface views tasks such as object localization, attribute comparison, or spatio-temporal analysis. To quantify this gap, we compared a specialist remote sensing model (GeoChat-7B(Kuckreja et al., 2023)) against LLaVA-1.5-7B on TDBench (details in Appendix E.2). While GeoChat excelled at detection tasks, it failed catastrophically on reasoning tasks, indicating that current remote sensing benchmarks do not cover the spatial reasoning required for near-surface aerial domains. Moreover, satellite images are typically captured from fixed nadir viewpoints at consistent altitudes, lacking the perspective variation and dynamic conditions common in drone operations.

Beyond remote sensing, only a few studies explore top-down images. For instance, Li et al. (2024a) introduces an indoor map benchmark for evaluating navigation and spatial reasoning from floor plans. In contrast, our benchmark TDBench focuses on high-resolution, near-surface top-down images resembling drone viewpoints, enabling systematic evaluation of fine-grained perception and reasoning abilities that remain underrepresented in existing benchmarks.

### 2.3 HALLUCINATIONS IN MULTIMODAL LLMS

Hallucination has become an increasing concern in both large language models (LLMs) and vision–language models (VLMs). In VLMs, it often occurs when models generate content that is inconsistent with the image, such as describing nonexistent objects, misrepresenting spatial relationships, or ignoring the visual input entirely (Wang et al., 2024). Recent studies have introduced benchmarks and methods to systematically evaluate these visual hallucinations. Li et al. (2023a) introduced the POPE method, which probes object hallucination by asking targeted presence/absence questions and measuring how often models falsely claim the existence of unseen objects. Liu et al.

**Figure 2: Proposed RotationalEval (RE) strategy.** In RE, each image is rotated three times to create four questions, with choices generated separately for each rotation. We illustrate a failure case in *object localization* where four choices align with four images, and the VLM answers three correctly but fails on one. 'GT' refers to ground truth.

(2024a) provided a large-scale study on hallucinations in VLMs and proposed automatic detection metrics based on grounding scores, which assess alignment between textual output and visual evidence. HallusionBench (Guan et al., 2024) proposed a benchmark designed to isolate hallucination behavior using paired, contrastive visual questions to reveal when models invent objects or attributes.

These approaches typically rely on comparing generated captions or answers against ground-truth annotations, using measures such as hallucination rate (percentage of fabricated objects), grounding accuracy (percentage of correctly grounded mentions), or contrastive consistency scores. However, current methods primarily treat hallucination as a binary outcome (hallucinated or correct) and do not assess whether correct answers arise from genuine visual understanding or from chance agreement with priors. Our benchmark TDBench complements these efforts by a reliability-oriented evaluation perspective, aiming to distinguish reliably grounded responses from lucky successes.

## 3 DESIGN OF TDBENCH

In this section, we provide a brief overview of TDBench. More details regarding question examples, dataset implementation and quality control procedures are presented in Appendix B.

### 3.1 ABILITY TAXONOMY OF TDBENCH

TDBench evaluates top-down image understanding across **10 categories** derived from typical aerial tasks encountered in real-world applications. These categories span core aspects such as image perception, object identification, spatial reasoning, and multi-instance understanding as the dimensions shown in Figure 1 (Left). We excluded evaluation dimensions that are either common across existing benchmarks or largely unaffected by image perspective, such as text recognition or general knowledge recall, to focus the benchmark on perspective-sensitive capabilities.

### 3.2 DATA CONSTRUCTION

We constructed TDBench from two primary sources: curated public datasets (Shaha, 2025; Zhu et al., 2021; Gasienica-Jozkowy et al., 2021; ICG, 2019; Varga et al., 2022; Mou et al., in press) and realistic simulation (CARLA Simulator (Dosovitskiy et al., 2017) and GTA V). The benchmark includes two task types: Multiple Choice Questions (MCQs) for most abilities, and Visual Grounding (VG). Each MCQ problem is structured as a quadruple $P_i = [Q_i, I_i, C_i, L_i]$, where $Q_i$ denotes the textual question, $I_i$ is the associated image, $C_i$ represents the set of possible answers with $n$ ($2 \leq n \leq 4$) choices $\{c_1, c_2, \ldots, c_n\}$ (randomly shuffled during evaluation), and $L_i$ is the correct label. For VG problems, we evaluate models' ability to precisely localize objects by comparing their predicted bounding box coordinates against $L_i$, which contains human-annotated ground truth coordinates. In addition, all input images in TDBench are standardized to a square resolution of 512×512 pixels to eliminate variability from model-specific preprocessing, which could otherwise affect the results.

### 3.3 LEVERAGING ROTATIONAL INVARIANCE IN EVALUATION

In TDBench, we introduce a novel evaluation strategy, **RotationalEval (RE)**, designed to leverage the unique properties of top-down images (Figure 2, example from *object localization*). RE evaluates

model performance on four orientations of each image: the original, 90°, 180°, and 270° rotations, and counts a question as correct only if **all** four are answered correctly. This exploits the fundamental **rotational invariance** of aerial perspectives. Unlike front-view images, where rotations create physically implausible scenes, top-down rotations simply mimic different yaw angles without altering scene content. During evaluation, we treat each rotation as a stateless, independent instance. Both the image and text prompts (e.g., directional references) are distinct for each orientation, the model cannot exploit correlations between trials.

## 3.4 TDBench Statistics

TDBench contains 2000 problems across the 10 ability categories for each rotation, plus an additional 2100 problems used in four case studies. We aimed for an even distribution of problems across abilities, with 200 samples per category. Of the total questions, 1910 (including case studies) are collected from real-world datasets, and 2190 are generated from simulation environments. Notably, all problems in the 'Object Counting' category are generated from the CARLA Simulator, which allows controlled ground-truth labeling during scene generation. Under RotationalEval (RE), each question is evaluated across four orientations, effectively producing four instances per problem.

## 4 EVALUATION RESULTS

### 4.1 SETUP

To ensure reproducibility and a fair comparison across models, all evaluations are conducted within an open-source VLM evaluation framework. We evaluated a total of 60 VLMs in a zero-shot setting, without providing any in-context examples. For all experiments, the model temperature was set to 0, and GPT-4o was used as the answer extractor for all model outputs.

**Models** We evaluated 17 **proprietary models**, including the Claude (Anthropic, 2024; 2025a;c;b), Gemini (Google, 2024; 2025a), and GPT (OpenAI, 2024; 2025a;c;b) families; and 43 **open-source models** from diverse families such as Gemma 3 (Google, 2025b), InternVL (Chen et al., 2025; Zhu et al., 2025; Wang et al., 2025), Qwen2.5-VL (Bai et al., 2025a), DeepSeek-VL2 (Zhiyu Wu, 2024), LLaVA (Liu et al., 2023; Li et al., 2024b), Kimi-VL (KimiTeam, 2025), and VLM-R1 (Shen et al., 2025). These models span a wide range of sizes, from 0.5 billion to 38 billion parameters.

Table 1: Performance comparison of open-source and proprietary VLMs under VanillaEval (VE@0°) and RotationalEval (RE), along with the corresponding accuracy drop ($\Delta$) on TDBench.

| Open VLMs | VE | RE | $\Delta$ | Prop VLMs | VE | RE | $\Delta$ |
|---|---|---|---|---|---|---|---|
| Qwen2.5-VL 7B | 0.630 | **0.470** | -0.160 | Gemini 2.5 Pro | **0.793** | **0.611** | -0.182 |
| Kimi-VL | 0.624 | 0.455 | -0.169 | Gemini 1.5 Pro | 0.756 | 0.572 | -0.183 |
| DeepSeek VL2 | **0.637** | 0.448 | -0.189 | GPT-5 | 0.761 | 0.570 | -0.190 |
| InternVL3.5 14B | 0.601 | 0.442 | -0.159 | GPT-4.1 | 0.720 | 0.520 | -0.200 |
| LLaVA-Next-13B | 0.617 | 0.419 | -0.198 | Claude Sonnet 3.7 | 0.611 | 0.415 | -0.196 |
| Gemma3 12B | 0.591 | 0.330 | -0.260 | Claude Opus 4.1 | 0.603 | 0.392 | -0.211 |

### 4.2 RESULTS

**RotationalEval vs. VanillaEval** We first compare our proposed RotationalEval (RE) with the conventional one-pass evaluation, VanillaEval (VE). Table 1 summarizes their results on TDBench, averaged across all dimensions. To validate the benchmark, we conducted a human study (excluding visual grounding), achieving 0.92 VE and 0.89 RE. The high accuracy confirms dataset solvability, while the minimal gap (0.03) validates RE as a consistent metric for genuine understanding.[1] Adopting RE leads to a notable performance decline across all VLMs. This drop occurs because RE reduces the chance of obtaining correct answers through random guessing. Interestingly, models with higher

---

[1]In contrast, text-only model baselines yielded a VE of $\approx 33\%$ (close to the random guess baseline of 30.6%), confirming that TDBench requires visual reasoning and cannot be solved via language priors.

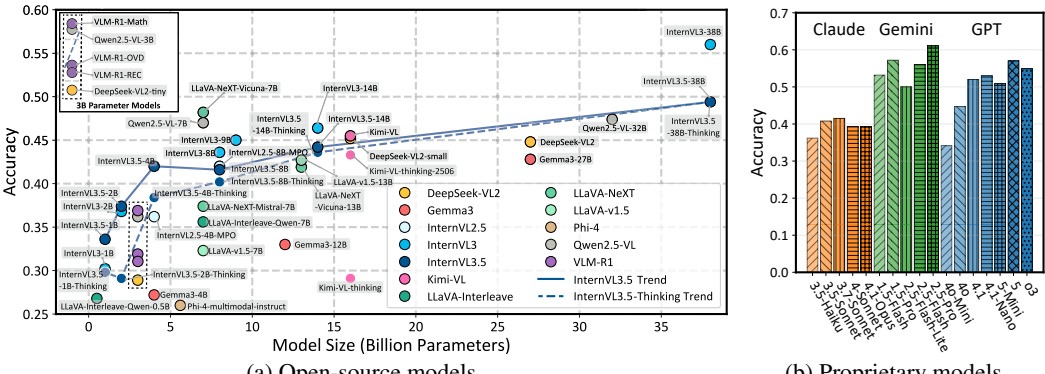

(a) Open-source models.  (b) Proprietary models.

Figure 3: Average RE performance of models on TDBench, aggregated across 10 evaluation dimensions for both Open-source and Proprietary models.

VE do not necessarily achieve higher RE. For example, although Gemini 1.5 Pro has a slightly lower VE than GPT-5 (0.756 vs. 0.761), it attains a higher RE (0.572 vs. 0.570). Among open models, DeepSeek VL2 achieves the best VE, while Qwen2.5-VL-7B achieves the highest RE. These results suggest that models performing well under VE may still be prone to hallucinations, which we further examine in Section 4.3.

**Main Results**   All reported results are based on **RotationalEval (RE)**, calculated as the *average* across ten evaluation categories unless explicitly stated. Detailed results, including *dimension-wise performance*, are provided in Appendix E. Figure 3a shows the RE performance of various open-source models as a function of their parameter size. Within the same model families, performance generally increases with model size, although several exceptions exist. Notably, the *"thinking" variants consistently underperform their standard counterparts*, especially at smaller model sizes, with the gap narrowing as model size increases. This suggests that while chain-of-thought prompting can enhance reasoning at the semantic level, it may make responses less grounded in the visual input. In addition, newer models do not necessarily perform better: for example, InternVL3.5 underperforms InternVL3 despite being trained on more data, suggesting that additional general-purpose data may have diluted the proportion of top-down-related images during training. We also report the performance of proprietary models in Figure 3b; although their parameter sizes are undisclosed, the largest variants generally outperform their smaller counterparts, except for GPT-4.1 and GPT-4.1-Nano.

## 4.3 BEYOND ACCURACY: A DEEPER ANALYSIS OF MODEL RELIABILITY

As noted earlier, RotationalEval (RE) yields lower scores than VanillaEval (VE) because it discounts isolated correct predictions and thus reduces the impact of lucky guesses. To further analyze this

Table 2: RE, MA, $\overline{\text{VE}}$, and reliability parameters (proportion of questions a model truly knows $\theta$, accuracy among known questions $r$, accuracy among guessed questions $g$, and adjusted accuracy $A_{\text{adj}}$). Arrows indicate whether higher ($\uparrow$) or lower ($\downarrow$) is better. Best values are green, worst are red.

| **Model** | **RE**$\uparrow$ | **MA**$\downarrow$ | $\overline{\textbf{VE}}\uparrow$ | $\theta\uparrow$ | **r**$\uparrow$ | **g** | $A_{\textbf{adj}}\uparrow$ |
|---|---|---|---|---|---|---|---|
| Gemini 2.5 Pro | **0.611** | **0.073** | **0.791** | **0.822** | 0.909 | 0.201 | **0.754** |
| GPT-5 | 0.570 | 0.085 | 0.751 | 0.688 | **0.941** | 0.265 | 0.652 |
| Claude Opus 4.1 | **0.392** | 0.194 | 0.607 | 0.610 | **0.849** | 0.189 | 0.541 |
| o3 | 0.549 | 0.096 | 0.731 | 0.693 | 0.921 | 0.279 | 0.651 |
| DeepSeek VL2 | 0.448 | 0.196 | 0.631 | 0.620 | 0.900 | 0.184 | 0.568 |
| Gemma3-27B | 0.428 | 0.220 | **0.604** | **0.587** | 0.880 | 0.206 | **0.538** |
| Qwen2.5-VL-32B | 0.474 | 0.165 | 0.668 | 0.668 | 0.902 | 0.203 | 0.611 |
| Kimi-VL | 0.455 | **0.239** | 0.613 | 0.612 | 0.882 | 0.164 | 0.565 |

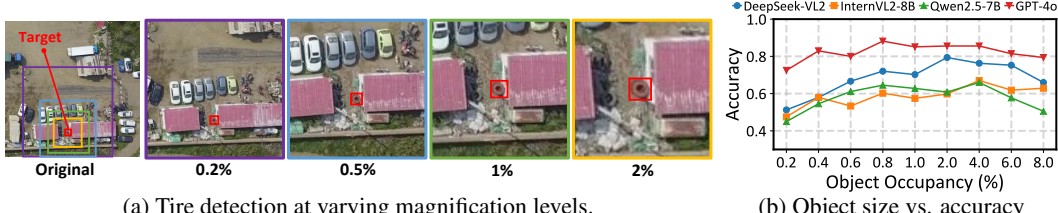

(a) Tire detection at varying magnification levels.    (b) Object size vs. accuracy

Figure 4: Impact of digital magnification on aerial object detection performance.

phenomenon, let $\Phi = \{0°, 90°, 180°, 270°\}$ be the set of rotations, and let $Y_i^{(\phi)} \in \{0,1\}$ denote whether the question $i$ under rotation $\phi$ is answered correctly. We define three observations

$$\mathrm{RE} = \Pr(\forall \phi \in \Phi : Y_i^{(\phi)} = 1), \quad \overline{\mathrm{VE}} = \mathbb{E}\Big[\frac{1}{|\Phi|}\sum_{\phi \in \Phi} Y_i^{(\phi)}\Big], \quad \mathrm{MA} = \Pr(\forall \phi : Y_i^{(\phi)} = 0),$$

where MA denotes wrong answer in all rotations. Assuming each question for the model is either "known" or "unknown", and rotations are conditionally independent, the above observations satisfies

$$\mathrm{RE} = \theta r^4 + (1-\theta)g^4, \qquad \overline{\mathrm{VE}} = \theta r + (1-\theta)g, \qquad \mathrm{MA} = \theta(1-r)^4 + (1-\theta)(1-g)^4.$$

where $\theta$ represents *the proportion of questions the model truly knows*, $r$ means *the accuracy on known questions*, and $g$ denotes *the accuracy on unknown questions* (due to lucky guesses). These parameters are inferred by solving the system of equations above (see Appendix C for derivation); We aggregate these into the adjusted accuracy ($A_{\mathrm{adj}}$):

$$A_{\mathrm{adj}} = \theta \cdot r.$$

The adjusted accuracy represents single-pass accuracy after discounting the contribution of guessing from the apparent correctness (VE). To illustrate this, Figure 1 (Right) presents results for different sizes of InternVL3, averaged across all evaluation dimensions. As model size increases, $\theta$ (the proportion of questions the model truly knows) also rises, while $r$ remains consistently high (approaching 100%), and shows a gradual upward trend with scale, which is desirable. In contrast, $g$ exhibits variability that does not show a clear dependence on model size. Overall, Adjusted Accuracy improves with larger models, supporting the validity of our probability-based knowledge reliability analysis. Table 2 reports additional representative results across different model families (including four proprietary and four open-source VLMs), with full category-wise breakdowns for all 60 models provided in Appendix E.

Unlike the scaling trend observed with InternVL3, different models exhibit distinct strengths and weaknesses on TDBench. For example, *Gemini 2.5 Pro* achieves the highest $\theta$, suggesting it possesses the broadest knowledge coverage, although its $r$ is lower than that of OpenAI's GPT-5 and o3. Both GPT-5 and o3, however, yield the highest $g$ values, indicating that these models are more likely to produce correct answers by chance. On the other hand, *Gemma3-27B* shows the lowest $\theta$, indicating a comparatively narrower knowledge base. Meanwhile, *Claude Opus 4.1* shows the lowest $r$ among all models, even below all open-source models listed here, which may stem from its stronger emphasis on code-related reasoning or function-calling tasks rather than visual–language understanding.

**Probing Intrinsic Model Properties**    Although we introduce these metrics within the context of TDBench, they are not inherently tied to top-down image understanding. Rather, TDBench serves as a probing medium to reveal latent aspects of model behavior that cannot be directly observed. The estimated parameters $(\theta, r, g)$ reflect how much of a model's correctness stems from genuine knowledge versus lucky guesses, capturing properties intrinsic to the model itself rather than any particular dataset.

## 5   CASE STUDIES

Top-down images are typically captured from high altitudes, which introduces unique challenges such as small object size, unusual perspective, and the lack of depth cues, yet depth is critical for tasks like building height estimation or drone navigation. To examine these challenges, we design four targeted case studies.

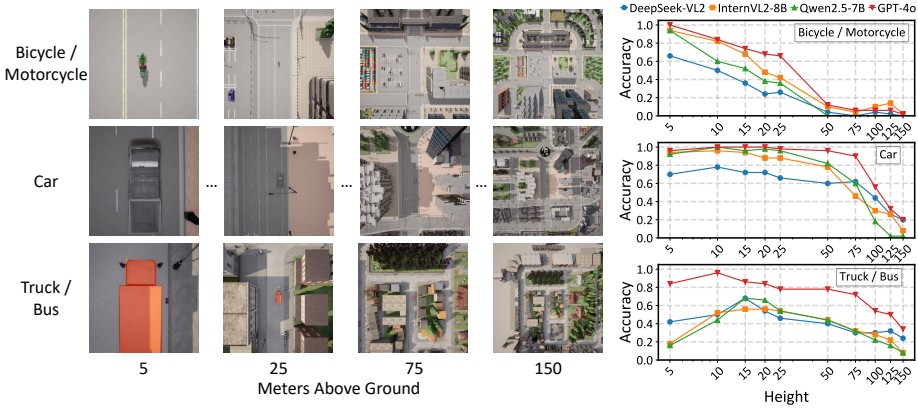

(a) CALRA Simulation for multi-altitude image capture.  (b) Height vs. accuracy.

Figure 5: Impact of camera altitude on object detection performance. The right plot shows detection accuracy as a function of altitude (5-150m) on a logarithmic scale.

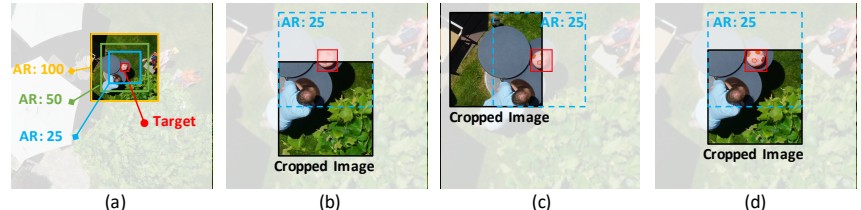

Figure 6: Example of Integrity study. (a) Three different area ratios (AR) ($25\times$, $50\times$, $100\times$). (b), (c), and (d) show visibility ratios of 30%, 60%, and 90%, respectively, in the setting (AR=$25\times$), depending on how much of the object is bounded inside the image cropped regions.

## 5.1 CASE STUDY 1: DIGITAL MAGNIFICATION FOR SMALL OBJECT DETECTION

Small objects occupy very few pixels, making them difficult for VLMs to detect. We explore a *digital magnification* strategy that crops images to increase the target object's relative pixel coverage (area ratio), as illustrated in Figure 4a. We use samples from *object presence* and *object localization* tasks where baseline performance was low, and reformat them using the *object presence* template.

Figure 4b shows that accuracy rises with area ratio before dropping as context is lost. GPT-4o peaks at only 0.8% occupancy, whereas open-source models require 2–4%. Beyond 6%, performance declines across all models due to resolution loss and reduced context. These findings offer practical guidance on magnification levels for aerial imaging and suggest future work on improving small-object detection in VLMs, particularly for models using multi-tile preprocessing, where tile size could be adapted based on prior knowledge of target object scale.

## 5.2 CASE STUDY 2: ALTITUDE EFFECTS ON OBJECT DETECTION

This study examines optimal hovering heights for drones with a fixed field of view (FOV) when performing tasks that require consistent object detection, such as tracking suspects. Unlike previous studies, we focus on physical "zoom-in", where the drone adjusts its altitude to improve detection performance. Because most datasets lack camera height metadata, we used the CARLA simulation to deploy multiple cameras at different altitudes over identical scenes (Figure 5a). We evaluated three object categories (bicycle/motorcycle, car, and truck/bus—chosen) for their frequency in aerial tasks and distinct size differences. *Object presence performance* was measured across altitudes from 5 to 150 meters, spanning typical operational ranges for commercial and tactical drones, while keeping image resolution constant. This setup offers practical guidance for maximizing detection reliability through optimal drone positioning rather than post-capture image processing.

As shown in Figure 5b, accuracy generally decreases with altitude but peaks at specific heights: 5m for bicycles/motorcycles, 10m for cars, and 15m for trucks/buses. We attribute this to field coverage

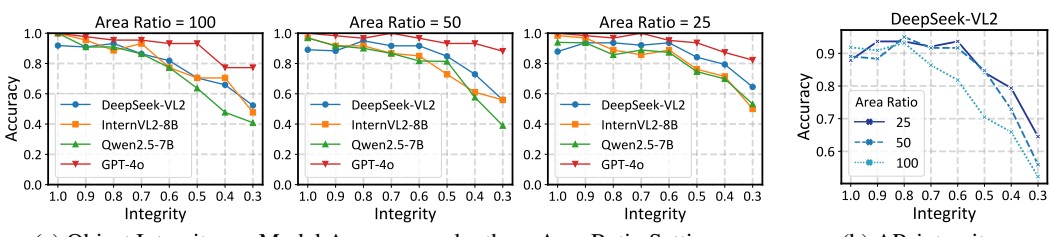

(a) Object Integrity vs. Model Accuracy under three Area Ratio Settings.

(b) AR-integrity.

Figure 7: Impact of Object Integrity and Area Ratio on VLM Performance.

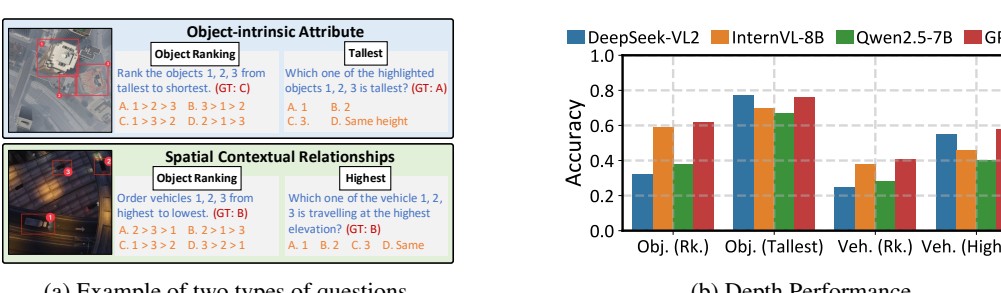

(a) Example of two types of questions.

(b) Depth Performance.

Figure 8: Analysis of spatial awareness and depth perception.

differences: at low altitudes, large objects may be only partially visible, reducing detection accuracy, while smaller objects remain fully visible even at minimal heights.

## 5.3 Case Study 3: Object Visibility and Partial Occlusion

Objects may be only partially visible, especially near image borders. We controlled visibility (**integrity**) by shifting a fixed-size crop window over objects at a set area ratio (AR) (Figure 6). This allowed us to vary integrity while keeping magnification constant.

Figure 7a shows that accuracy stays stable ($\geq$90%) until integrity drops below a threshold, then declines sharply. This threshold depends on AR: with AR=100, accuracy drops below 70% integrity, while lower ARs fail around 60%(Figure 7b). This demonstrates how incomplete visibility affects detection even without resolution changes.

## 5.4 Case Study 4: Z-Axis Perception and Depth Understanding

Since top-down images preserve xy-plane information, they inherently lack altitude cues. To evaluate this limitation, we defined two types of **z-axis awareness** challenges (Figure 8a): (i) assessing an object's intrinsic properties, such as a building's or tree's height, and (ii) evaluating contextual relationships, such as determining whether a car is traveling on a road or an overpass. As shown in Figure 8b, DeepSeek performs well on tallest/highest identification but struggles with ranking tasks, whereas GPT-4o achieves near-best performance across both types.

## 6 Conclusion

In this work, we introduced TDBench, a comprehensive benchmark for evaluating VLMs on top-down images, comprising over 2,000 manually labeled questions across diverse categories. To ensure robust and reliable assessment, we proposed **RotationalEval**, an evaluation strategy that leverages the rotational invariance of top-down perspectives to provide a more rigorous alternative to standard single-pass evaluation. Beyond accuracy, we further developed a set of **reliability-oriented metrics** that assess how much of a model's performance stems from genuine knowledge rather than lucky guesses or hallucinated responses. Our multi-dimensional analysis reveals both the capabilities and limitations of current VLMs, and our four case studies demonstrate their strengths and challenges in real-world aerial applications. While TDBench serves as the testbed for this study, these metrics are not tied to any specific dataset and can serve as **general probes of model reliability**, offering a new perspective for guiding future development of more trustworthy VLMs.

## 7 ETHICS STATEMENT

We acknowledge the potential use of TDBench in areas such as automated surveillance and military systems. While our goal is to promote positive applications like civilian navigation and environmental monitoring, we mitigate these risks through open research and restrictive licensing. TDBench is released as a public benchmark under the CC BY-NC-SA 3.0 IGO license, which restricts commercial use, discouraging deployment in for-profit surveillance or military settings. The benchmark is built from public and simulated data, and we encourage its responsible use.

## 8 REPRODUCIBILITY STATEMENT

All experiments in this manuscript were conducted using an open-source evaluation framework, with TDBench designed for full compatibility. We will release the evaluation code and detailed commands upon publication. Due to storage limitations, raw model outputs are not included in the supplementary material; instead, we provide aggregated results that allow reproduction of the reported results. These include the main results in Tables 1 and 2, as well as the detailed results in Tables 6–9 in the appendix.

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

# Appendix

## A  LARGE LANGUAGE MODEL USAGE ACKNOWLEDGMENT

We used large language models (LLMs) to assist in the preparation of this work in the following ways. First, LLMs were employed for language-related support, including polishing the writing and improving grammar, clarity, and overall readability of the manuscript. Second, LLMs were used as coding assistants primarily for generating and refining code to produce figures for the paper. All research ideas, experimental designs, analyses, and final claims presented in this work were conceived, validated, and verified by the authors. The authors take full responsibility for the content of this paper.

## B  MORE DETAILS ABOUT THE TDBENCH

### B.1  BENCHMARK TAXONOMY

In this section, we provide an overview of the 10 categories in TDBench with examples in Figure 9. We then describe the data sources used to build the benchmark and the procedures for curating and annotating the dataset.

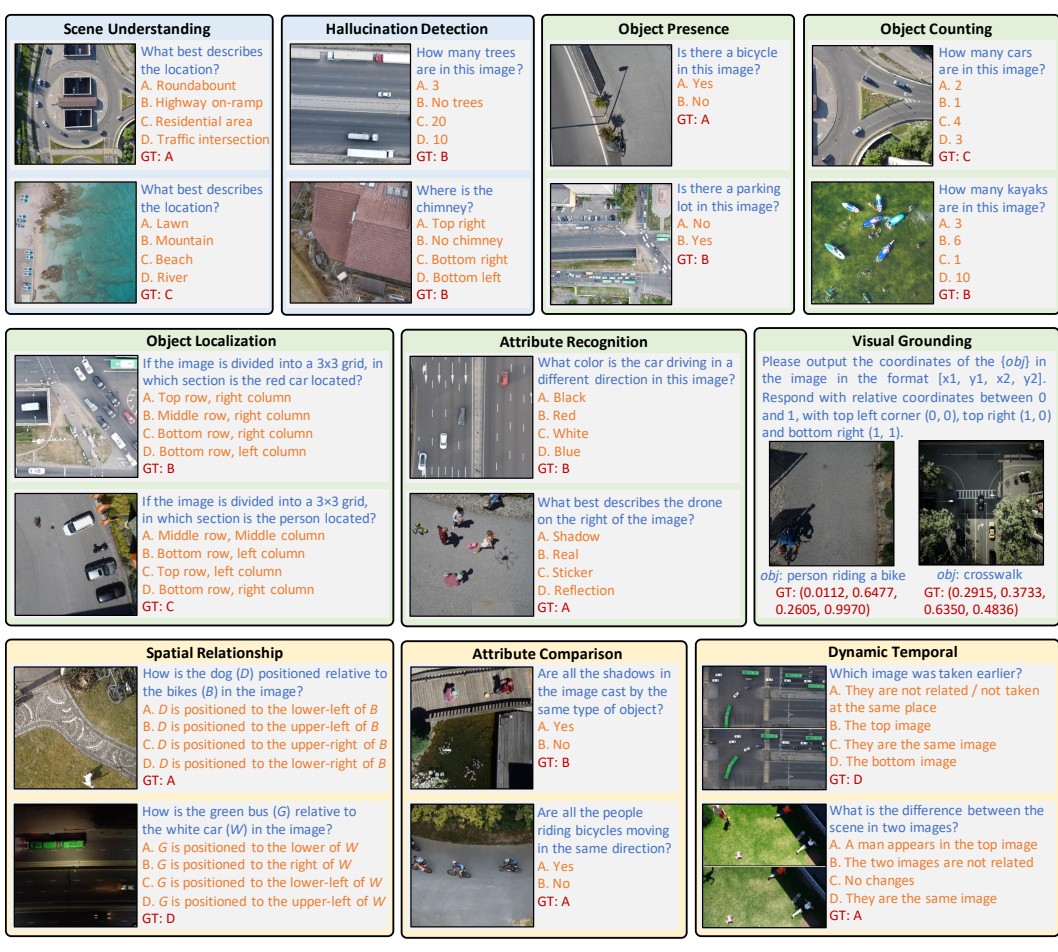

Figure 9: Benchmark examples across the ten categories in TDBench. Different colors indicate the three high-level capability groups: image perception (blue), single-instance understanding (green), and multi-instance reasoning (yellow).

**Image Perception** This category focuses on the broad-scale interpretation of top-down aerial imagery, emphasizing holistic semantic understanding rather than fine-grained details. Such capabilities are especially valuable for wide-area reconnaissance, where drones must scan large regions to detect critical features such as wildfire outbreaks, traffic congestion, or emergency response scenarios. It includes two tasks: *Scene Understanding*, which evaluates a model's ability to comprehend the overall contextual meaning of a scene, and *Hallucination Detection*, which assesses its ability to distinguish actual image content from fabricated choices. These tasks are shown in **blue** in Figure 9 and represent foundational abilities for reliable aerial image interpretation.

**Single-Instance Understanding** This category emphasizes detailed object-level recognition and localization within a single image, as shown in **green** in Figure 9. It covers both recognition and localization aspects. For recognition, *Object Presence* evaluates basic detection capabilities, and *Attribute Recognition* assesses the identification of specific properties such as color, shape, material, or species. For localization, we use a three-tiered approach: coarse presence detection (*Object Presence*), intermediate 3×3 grid-based localization (*Object Localization*) requiring quadrant-level precision, and fine-grained *Visual Grounding* using exact bounding box coordinates. We also include *Object Counting* to assess quantification abilities, which is particularly challenging in aerial contexts where many similar objects appear at varying scales and densities.

**Multi-Instance Reasoning** This category evaluates compositional reasoning across multiple objects, requiring analysis of spatial, comparative, and temporal relationships, as shown in **yellow** in Figure 9. *Spatial Relationship* tasks challenge models to localize multiple objects and accurately determine their relative positions, which is crucial for navigation and path planning in autonomous aerial systems. *Attribute Comparison* requires models to compare properties or states across multiple entities, useful for anomaly detection and identifying distinctive features. Finally, *Dynamic Temporal* presents pairs of images to evaluate models' ability to detect changes, reason about temporal order, and infer causal relationships.

## B.2 DATA SOURCES

To maximize data diversity, we combined multiple open-source datasets covering varied environments, including urban infrastructure, remote wilderness, and disaster zones (Table 3). All images from these datasets were manually selected and annotated following our evaluation taxonomy. In addition to real-world data, we generated synthetic images using the CARLA simulator with custom scripts to control scene parameters precisely. For specialized case studies requiring exact ground truth, such as camera altitude, object counts, or height measurements, we used both CARLA and *Grand Theft Auto V (GTA V)*.

Table 3: Distribution of data sources in TDBench

| Image Source | Problem Formulation | **Number** | Ratio |
|---|---|---|---|
| Aerial Traffic Images (Shaha, 2025) | Human Annotation | **457** | 20.8% |
| Semantic Drone (ICG, 2019) | Human Annotation | **653** | 29.7% |
| AFO (Gasienica-Jozkowy et al., 2021) | Human Annotation | **18** | 0.8% |
| Visdrone (Zhu et al., 2021) | Human Annotation | **416** | 18.9% |
| Seadronesee (Varga et al., 2022) | Human Annotation | **3** | 0.1% |
| ERA (Mou et al., in press) | Human Annotation | **363** | 16.5% |
| CARLA (Dosovitskiy et al., 2017) | Simulation Script | **290** | 13.2% |
| **Additional New Data Used In Case Study** | | | |
| CARLA (Dosovitskiy et al., 2017) | Simulation Script | **1500** | - |
| GTA V | Human Annotation | **400** | - |

## B.3 IMPLEMENTATION OF TDBENCH

**Rotation-Aware Question Design** Because TDBench supports RotationalEval (RE), we categorized all questions as either **rotation-invariant** or **rotation-sensitive**. Rotation-invariant questions (e.g., object presence, attribute recognition) remain semantically unchanged after rotation; only the image is rotated while the question and answer options remain the same. Rotation-sensitive

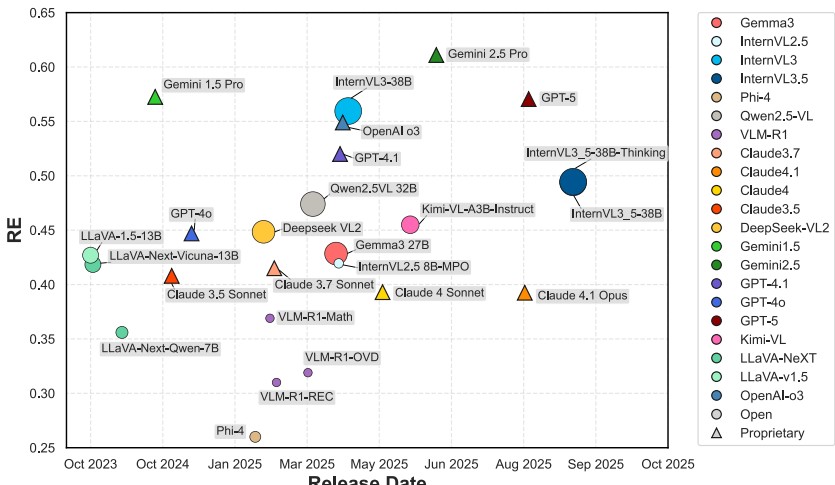

Figure 10: Performance (RE) of models versus their release date. Circles denote open-source models, with marker size indicating model scale. Triangles denote proprietary models. Each point represents the largest evaluated model from a given family.

questions (e.g., spatial relationships or localization) require synchronized transformation of directional references. For instance, after a 90° clockwise rotation, phrases like "top-left" are mapped to "top-right".

To automate this process, we use placeholder tokens ($\langle\text{img1}\rangle$, $\langle\text{img2}\rangle$) in both questions and answers. In the original orientation, they are rendered as "left/right" or "top/bottom", and these tokens are automatically rotated when generating the 90°, 180°, and 270° variants. This ensures consistent semantics across all rotation conditions.

**Image Standardization** To mitigate evaluation biases from inconsistent image preprocessing across different VLMs (such as padding, stretching, or multi-tiling), we established a uniform input pipeline. All images were standardized to a fixed 512×512 pixel resolution. For tasks requiring image pairs, such as temporal or comparative analyses, we concatenated two sub-images either horizontally (as a 512×256 pair) or vertically (as a 256×512 pair). This method ensures the combined input fits the same 512×512 canvas, providing a fair and consistent basis for model comparison.

**Quality Control** We followed a two-stage quality control pipeline combining human and model-based checks. *Stage 1: Human review.* Six annotators independently examined all questions, removing or revising items that were unsolvable due to lost context during cropping, or that contained unclear wording or incorrect ground truth. *Stage 2: Model filtering.* Several open-source models were benchmarked to detect consistently failed or consistently solved items. Questions that all models failed underwent additional human review and were retained only if correctly formulated, while those that all models solved were discarded for offering little discriminative value in model comparison.

**CARLA Simulation** CARLA (Dosovitskiy et al., 2017) is an open-source autonomous driving simulator that provides high-fidelity urban environments and physics. We used its configurable RGB and segmentation cameras at various altitudes to generate synthetic data. This setup enables precise control over object instances (e.g., vehicles), supporting systematic evaluation of object counting performance (Section 3) and altitude-dependent detection studies (Section 5).

## C IDENTIFIABILITY OF THE MIXTURE PARAMETERS

We used three parameters, $(\theta, r, g) \in [0, 1]^3$ to study the reliability of the models. These parameters denote the proportion of questions a model truly knows ($\theta$), model's accuracy among the questions that it knows ($r$), and model's accuracy among the questions it does not know and guessed ($g$).

We show that these parameters in our mixture model are *generically unique* given the observed statistics

$$\text{RE}, \qquad \overline{\text{VE}}, \qquad \text{MA}.$$

## C.1   PROBLEM FORMULATION

Assume the parameters satisfy

$$\begin{aligned}
\text{RE} &= \theta r^4 + (1 - \theta)g^4, \\
\overline{\text{VE}} &= \theta r + (1 - \theta)g, \\
\text{MA} &= \theta(1 - r)^4 + (1 - \theta)(1 - g)^4.
\end{aligned} \tag{1}$$

This system is symmetric under the transformation

$$(\theta, r, g) \longleftrightarrow (1 - \theta, g, r).$$

To remove this trivial multiplicity, we restrict to the *ordered domain*

$$\mathcal{D}_{\overline{\text{VE}}} := \{(r, g) \mid 0 \leq g < \overline{\text{VE}} < r \leq 1\}, \qquad \theta = \frac{\overline{\text{VE}} - g}{r - g} \in (0, 1). \tag{2}$$

The degenerate case $r = g$ occurs iff $\text{RE} = \overline{\text{VE}}^4$ and $\text{MA} = (1 - \overline{\text{VE}})^4$ and is excluded. We also exclude trivial boundary cases $\overline{\text{VE}} \in \{0, 1\}$ or $\text{MA} \in \{0, 1\}$, where conditioning becomes ill-defined.

## C.2   REDUCTION TO SECANT EQUATIONS

Define $f(x) = x^4$ and $u(x) = (1 - x)^4$. Eliminating $\theta$ using the middle equation in equation 1, the outer equations become the *secant identities*

$$\boxed{\frac{\text{RE} - g^4}{\overline{\text{VE}} - g} = \frac{r^4 - g^4}{r - g}, \qquad \frac{\text{MA} - (1 - g)^4}{\overline{\text{VE}} - g} = \frac{(1 - r)^4 - (1 - g)^4}{r - g}}. \tag{3}$$

These state that $(g, r)$ have the same secant slope on $f$ as $(1 - g, 1 - r)$ do on $u$.

Since $f$ and $u$ are strictly convex on $[0, 1]$, their secant slopes are strictly increasing in each endpoint. In particular, for fixed $g \in [0, 1)$,

$$r \mapsto \frac{r^4 - g^4}{r - g} \quad \text{is strictly increasing on } (g, 1]. \tag{4}$$

## C.3   ELIMINATION TO ONE VARIABLE

Using $r^4 - g^4 = (r - g)(r^3 + gr^2 + g^2 r + g^3)$, the first equation in equation 3 is equivalent (for $r \neq g$) to the cubic

$$r^3 + gr^2 + g^2 r + g^3 = \frac{\text{RE} - g^4}{\overline{\text{VE}} - g}. \tag{5}$$

By equation 4, this has at most one solution $r > g$ for each fixed $g$.

Let $r = R(g)$ denote this unique solution (if it exists) and define

$$E(g) := \frac{(1 - R(g))^4 - (1 - g)^4}{R(g) - g} - \frac{\text{MA} - (1 - g)^4}{\overline{\text{VE}} - g}. \tag{6}$$

**Lemma (Bijection with $E(g) = 0$).**   *For fixed $\overline{\text{VE}} \in (0, 1)$, ordered solutions $(r, g) \in \mathcal{D}_{\overline{\text{VE}}}$ of equation 3 are in one-to-one correspondence with real roots $g \in (0, \overline{\text{VE}})$ of $E(g) = 0$ for which $R(g) > \overline{\text{VE}}$. For each such root $g^\star$, the corresponding $r^\star = R(g^\star)$ is unique, and then $\theta^\star = \frac{\overline{\text{VE}} - g^\star}{r^\star - g^\star}$.*

**Proof.** Fix $g \in (0, \overline{\text{VE}})$. The first equality in equation 3 uniquely determines $r = R(g) > g$ by equation 5; substituting into the second gives $E(g) = 0$. Conversely, if $E(g) = 0$ and $R(g) > \overline{\text{VE}}$, then $(\theta, r, g) = (\frac{\overline{\text{VE}} - g}{R(g) - g}, R(g), g)$ solves equation 1.   $\square$

**Remark (Why $R(g) > \overline{\mathrm{VE}}$).** Since $x \mapsto x^4$ is convex on $[0, 1]$, Jensen's inequality gives $\mathrm{RE} = \theta r^4 + (1-\theta)g^4 \geq (\theta r + (1-\theta)g)^4 = \overline{\mathrm{VE}}^4$, with strict inequality in the ordered, nondegenerate case $r \neq g$. Hence

$$\frac{\mathrm{RE} - g^4}{\overline{\mathrm{VE}} - g} \;>\; \frac{\overline{\mathrm{VE}}^4 - g^4}{\overline{\mathrm{VE}} - g}.$$

By strict monotonicity in equation 4, the unique $r$ satisfying the first secant identity must satisfy $r = R(g) > \overline{\mathrm{VE}}$.

## C.4 THE CUBIC IN $g$ AND ITS DISCRIMINANT

Clearing denominators in equation 3 yields a cubic polynomial

$$P_{\mathrm{RE}, \overline{\mathrm{VE}}, \mathrm{MA}}(g) \;=\; 0, \tag{7}$$

whose coefficients depend algebraically on $(\mathrm{RE}, \overline{\mathrm{VE}}, \mathrm{MA})$. *Degree justification.* Using equation 5, the first secant identity expresses $\frac{r^4 - g^4}{r - g}$ as $r^3 + gr^2 + g^2 r + g^3$, which is linear in the unknown slope $\frac{\mathrm{RE} - g^4}{\overline{\mathrm{VE}} - g}$; substituting this $r = R(g)$ into the second identity and clearing denominators cancels the factor $(r - g)$ and leaves a polynomial of degree at most 3 in $g$. (Explicit coefficients are lengthy and omitted for brevity.)

By the lemma above, *ordered solutions* are in bijection with *real roots of $P_{\mathrm{RE}, \overline{\mathrm{VE}}, \mathrm{MA}}(g)$ in $(0, \overline{\mathrm{VE}})$*.

Let $\Delta(P)$ denote the discriminant of a cubic $P(g) = ag^3 + bg^2 + cg + d$:

$$\Delta(P) = 18abcd - 4b^3 d + b^2 c^2 - 4ac^3 - 27a^2 d^2.$$

This determines the real root structure:

$$\Delta < 0 \;\Rightarrow\; \text{one real root}, \qquad \Delta > 0 \;\Rightarrow\; \text{three real roots}, \qquad \Delta = 0 \;\Rightarrow\; \text{a multiple real root}.$$

**Theorem (Uniqueness certificate).** *Fix $\overline{\mathrm{VE}} \in (0, 1)$ and $(\mathrm{RE}, \mathrm{MA})$. Let $P_{\mathrm{RE}, \overline{\mathrm{VE}}, \mathrm{MA}}$ be as in equation 7. If $\Delta(P_{\mathrm{RE}, \overline{\mathrm{VE}}, \mathrm{MA}}) < 0$, then there is at most one ordered solution $(r, g) \in \mathcal{D}_{\overline{\mathrm{VE}}}$. If, in addition, $P_{\mathrm{RE}, \overline{\mathrm{VE}}, \mathrm{MA}}$ has a real root $g^\star \in (0, \overline{\mathrm{VE}})$, then*

$$r^\star = R(g^\star), \qquad \theta^\star = \frac{\overline{\mathrm{VE}} - g^\star}{r^\star - g^\star}$$

*gives the unique solution $(\theta^\star, r^\star, g^\star)$ of equation 1 up to symmetry.*

**Proof.** Ordered solutions correspond to real roots of $P_{\mathrm{RE}, \overline{\mathrm{VE}}, \mathrm{MA}}(g)$ in $(0, \overline{\mathrm{VE}})$. If $\Delta < 0$ then $P$ has a single real root on $\mathbb{R}$, hence at most one in $(0, \overline{\mathrm{VE}})$. If such a root exists, the corresponding $(r, g)$ and $\theta$ are uniquely recovered via $R(g)$ and equation 2. $\qquad\square$

## C.5 GENERIC UNIQUENESS AND THE DISCRIMINANT LOCUS

Let $\mathcal{R}_{\overline{\mathrm{VE}}}$ be the image of $\mathcal{D}_{\overline{\mathrm{VE}}}$ under the map $(r, g) \mapsto (\mathrm{RE}, \mathrm{MA})$ defined by equation 3. The equation $\Delta(P_{\mathrm{RE}, \overline{\mathrm{VE}}, \mathrm{MA}}) = 0$ defines a real algebraic curve $\Sigma_{\overline{\mathrm{VE}}} \subset \mathcal{R}_{\overline{\mathrm{VE}}}$ (the *discriminant locus*).

**Theorem (Generic uniqueness).** *For fixed $\overline{\mathrm{VE}} \in (0, 1)$:*

- *If $(\mathrm{RE}, \mathrm{MA}) \in \mathcal{R}_{\overline{\mathrm{VE}}} \setminus \Sigma_{\overline{\mathrm{VE}}}$, then $\Delta < 0$ and equation 1 has a unique solution $(\theta, r, g)$ up to symmetry.*
- *If $(\mathrm{RE}, \mathrm{MA}) \in \Sigma_{\overline{\mathrm{VE}}}$, then either a multiple solution occurs or three distinct solutions exist.*

*In particular, $\Sigma_{\overline{\mathrm{VE}}}$ has measure zero, so for almost all valid $(\mathrm{RE}, \overline{\mathrm{VE}}, \mathrm{MA})$ the parameters $(\theta, r, g)$ are uniquely identifiable up to symmetry.*

**Proof (sketch).** Off $\Sigma_{\overline{\mathrm{VE}}}$ the simple-root condition $(\partial P / \partial g) \neq 0$ holds generically; by continuity (implicit function theorem), the number of real roots is locally constant and equals 1, yielding a single ordered solution. On $\Sigma_{\overline{\mathrm{VE}}}$ the discriminant changes sign, creating a multiple or triple real root. $\quad\square$

**Existence note.** For statistics induced by any nondegenerate mixture in the ordered domain ($r > \overline{\mathrm{VE}} > g$), continuity of the forward map $(r, g) \mapsto (\mathrm{RE}, \mathrm{MA})$ and the intermediate value principle ensure that $P_{\mathrm{RE}, \overline{\mathrm{VE}}, \mathrm{MA}}(g)$ attains a real root in $(0, \overline{\mathrm{VE}})$. Empirically, all rows in our dataset satisfy this condition.

## C.6 SUMMARY

The system equation 1 admits at most three ordered solutions (six with symmetry). However, *generically* $\Delta < 0$, so there is exactly one ordered solution (and thus one $(\theta, r, g)$ up to symmetry). Empirically, our dataset lies in this generic region, which explains why the solver returns either zero or one solution per row.

## D DISTINCTION FROM MULTI-PASS EVALUATION AND MAJORITY VOTING.

Unlike multi-pass evaluation with majority voting, which evaluates output variability by repeatedly sampling responses for the same image–question pair, our RotationalEval (RE) framework assesses invariance under controlled changes in the visual input. In multi-pass evaluation, the image remains identical across trials and only the text side varies—models sample different responses from the same underlying probability distribution, and any divergence arises solely from stochastic decoding. Even when question order or answer choices are shuffled, these modifications occur entirely at the semantic level in text and do not alter the visual input to the model. By contrast, TDBench rotates the image and systematically updates the question text and spatial relations to match the new orientation. Each trial therefore presents a distinct visual configuration of the same scene, requiring the model to consistently ground its reasoning in the visual content rather than relying on language priors. This fundamental difference makes RE a measure of visual invariance and grounding, whereas multi-pass evaluation primarily measures response stability under repeated sampling.

## E ADDITIONAL EVALUATION RESULTS

### E.1 MODEL SCALING TRENDS

We further analyze model performance trends over time and model size. Figure 3a shows the relationship between RE performance and model size for various open-source models. To examine temporal trends, Figure 10 plots model performance against their release dates, where open-source models are shown as dots (with marker size indicating model scale) and proprietary models are shown as triangles. Overall, performance tends to rise with newer releases, particularly among proprietary models such as GPT-5 and Gemini 2.5 Pro. Open-source models also progress over time, though less consistently: for instance, InternVL3.5, released after InternVL3, shows no clear RE improvement despite comparable size. A similar pattern appears in the Claude family, where later models (e.g., Claude 4.1 Opus) underperform earlier Sonnet versions on RE. These patterns indicate that top-down visual understanding is not a prioritized objective in current training regimes; most models appear to focus on mainstream capabilities such as chat, long-context reasoning, or coding, while robustness on top-down views receives little explicit attention. This highlights the underexplored status of top-down images and the importance of benchmarks like TDBench that bring this gap into focus.

### E.2 COMPARISON WITH REMOTE SENSING MODEL

To investigate whether existing remote sensing VLMs can address TDBench, we compared **GeoChat-7B** (Kuckreja et al., 2023), a model fine-tuned on satellite imagery, against a generalist baseline, **LLaVA-1.5-7B**. As detailed in Table 4, GeoChat achieves perfect performance on *Object Presence* (RE 1.00), demonstrating robust detection capabilities for aerial views. However, it fails catastrophically on reasoning-intensive tasks, scoring 0.00 on *Spatial Relationship* and *Visual Grounding*, and significantly underperforming LLaVA on *Scene Understanding* (0.14 vs. 0.81). This performance dichotomy confirms that while fine-tuning on remote sensing data improves simple detection, it does not confer the fine-grained spatial logic and holistic scene reasoning required by TDBench, further validating the distinct domain gap between traditional satellite tasks and near-surface aerial understanding.

| Category | LLaVA-1.5-7B RE (VE) | GeoChat-7B RE (VE) |
|---|---|---|
| Object Presence | 0.32 (0.56) | **1.00 (1.00)** |
| Visual Grounding | **0.37 (0.67)** | 0.00 (0.01) |
| Scene Understanding | **0.81 (0.91)** | 0.14 (0.31) |
| Spatial Relationship | **0.04 (0.41)** | 0.00 (0.26) |
| Attribute Comparison | **0.64 (0.67)** | 0.59 (0.64) |
| Attribute Recognition | **0.39 (0.59)** | 0.14 (0.32) |
| Dynamic Temporal | 0.09 (**0.38**) | **0.16** (0.29) |
| Hallucination Detection | 0.28 (0.56) | **0.40 (0.63)** |
| Object Counting | **0.18 (0.36)** | 0.04 (0.26) |
| Object Localization | **0.12 (0.55)** | 0.10 (0.24) |

Table 4: Comparison between generalist (LLaVA) and remote sensing specialist (GeoChat) models on TDBench. GeoChat excels at detection but lacks spatial reasoning capabilities.

### E.3 DETECTION-ASSISTED PIPELINE ANALYSIS

To determine if a hybrid approach could surpass end-to-end VLMs, we implemented a detection-assisted pipeline. We fine-tuned a YOLOv11x model on a **subset** of TDBench data (achieving high robustness, VE=0.857, RE=0.791) and used it to pre-process images by **overlaying bounding boxes** on detected objects. These augmented images were then fed into GPT-4o. We compared this hybrid approach against the direct VLM baseline on two categories: *Object Localization* (161 questions) and *Object Presence* (133 questions). Results are summarized in Table 5. We observed two distinct behaviors. First, **detection aids existence**: for *Object Presence*, the pipeline significantly boosts performance, particularly on the subset where objects were successfully detected (Accuracy increases from 82.9% to 91.8%), as the explicit visual cue helps the VLM confirm object existence. Second, **visual aids hinder reasoning**: conversely, for *Object Localization*, rotational consistency (RE) drops from 0.292 to 0.248. Even when the object is correctly boxed (det_count > 0), the VLM is less accurate at placing it in the correct grid cell (0.671 vs 0.679). This suggests that overlaid boxes introduce visual clutter or bias the VLM's internal spatial mapping, disrupting the precise coordinate reasoning required for localization.

| Task | Metric | Detection-Augmented (YOLOv11x + GPT-4o) | Direct VLM (GPT-4o) |
|---|---|---|---|
| *Object Localization* (161 Qs × 4 rotations = 644 samples) | | | |
| All Samples | RE | 0.248 | **0.292** |
| | VE | **0.689** | 0.677 |
| *Subset* (det > 0) | Accuracy | 0.671 | **0.679** |
| *Object Presence* (133 Qs × 4 rotations = 532 samples) | | | |
| All Samples | RE | **0.647** | 0.609 |
| | VE | **0.835** | 0.782 |
| *Subset* (det > 0) | Accuracy | **0.918** | 0.829 |

Table 5: Performance comparison between a Detection-Augmented Pipeline and Direct VLM inference. While detection augmentation improves existence tasks, it degrades spatial consistency (RE) and localization accuracy.

### E.4 COMPREHENSIVE DIMENSION-WISE RESULTS

We have presented only aggregated performance summaries (Figure 1, Table 1 & 2) in previous sections. For completeness, Tables 6–9 provide the full dimension-wise results of all 60 evaluated models (17 proprietary and 47 open-source) across the 10 evaluation dimensions in TDBench. To verify that these dimensions capture distinct model capabilities rather than just generic quality, we computed Pearson correlations across the 60 models. The average correlation is weak ($\rho = 0.38$ for VE), confirming task diversity.

Table 6: VLMs in TDBench on Scene Understanding, Hallucination Detection, Object Presence.

| Model | Scene Understanding | | | | | | Hallucination Detection | | | | | | Object Presence | | | | | |
|---|---|---|---|---|---|---|---|---|---|---|---|---|---|---|---|---|---|---|
| | RE | VE | $\theta$ | $r$ | $g$ | $A_{adj}$ | RE | VE | $\theta$ | $r$ | $g$ | $A_{adj}$ | RE | VE | $\theta$ | $r$ | $g$ | $A_{adj}$ |
| *Proprietary VLMs* | | | | | | | | | | | | | | | | | | |
| Claude 3.5 Haiku | 0.740 | 0.864 | 0.853 | 0.965 | 0.278 | 0.823 | 0.835 | 0.901 | 0.884 | 0.986 | 0.260 | 0.871 | 0.390 | 0.601 | 0.627 | 0.888 | 0.119 | 0.557 |
| Claude 3.5 Sonnet | 0.775 | 0.899 | 0.896 | 0.964 | 0.338 | 0.863 | 0.635 | 0.828 | 0.855 | 0.928 | 0.234 | 0.794 | 0.430 | 0.650 | 0.640 | 0.905 | 0.197 | 0.579 |
| Claude 3.7 Sonnet | 0.780 | 0.892 | 0.861 | 0.975 | 0.384 | 0.839 | 0.745 | 0.881 | 0.907 | 0.952 | 0.189 | 0.864 | 0.325 | 0.537 | 0.530 | 0.885 | 0.146 | 0.469 |
| Claude 4 Sonnet | 0.745 | 0.865 | 0.885 | 0.958 | 0.150 | 0.848 | 0.600 | 0.796 | 0.772 | 0.938 | 0.315 | 0.724 | 0.330 | 0.534 | 0.525 | 0.890 | 0.140 | 0.467 |
| Claude 4.1 Opus | 0.800 | 0.899 | 0.896 | 0.972 | 0.266 | 0.871 | 0.515 | 0.743 | 0.742 | 0.912 | 0.254 | 0.677 | 0.340 | 0.550 | 0.526 | 0.896 | 0.165 | 0.472 |
| GPT 4o-mini | 0.870 | 0.934 | 0.940 | 0.981 | 0.197 | 0.922 | 0.745 | 0.875 | 0.846 | 0.968 | 0.365 | 0.819 | 0.465 | 0.635 | 0.642 | 0.923 | 0.120 | 0.592 |
| GPT-4o | 0.930 | 0.961 | - | - | - | - | 0.575 | 0.761 | 0.761 | 0.932 | 0.216 | 0.710 | 0.645 | 0.815 | 0.760 | 0.958 | 0.361 | 0.728 |
| GPT-4.1 Nano | 0.875 | 0.932 | 0.932 | 0.984 | 0.221 | 0.918 | 0.485 | 0.700 | 0.659 | 0.925 | 0.264 | 0.610 | 0.735 | 0.853 | 0.849 | 0.964 | 0.223 | 0.819 |
| GPT-4.1 | 0.915 | 0.961 | 0.943 | 0.992 | 0.456 | 0.935 | 0.405 | 0.629 | 0.642 | 0.891 | 0.158 | 0.572 | 0.725 | 0.855 | 0.848 | 0.961 | 0.263 | 0.815 |
| OpenAI o3 | 0.920 | 0.965 | 0.956 | 0.990 | 0.419 | 0.947 | 0.560 | 0.756 | 0.760 | 0.926 | 0.217 | 0.704 | 0.565 | 0.730 | 0.724 | 0.940 | 0.180 | 0.680 |
| GPT-5 mini | 0.920 | 0.949 | 0.951 | 0.992 | 0.116 | 0.943 | 0.185 | 0.388 | 0.431 | 0.809 | 0.068 | 0.349 | 0.865 | 0.919 | 0.906 | 0.988 | 0.249 | 0.895 |
| GPT-5 | 0.930 | 0.971 | 0.973 | 0.989 | 0.344 | 0.962 | 0.550 | 0.730 | 0.707 | 0.933 | 0.147 | 0.664 | 0.615 | 0.754 | 0.731 | 0.958 | 0.201 | 0.700 |
| Gemini 1.5 Flash | 0.905 | 0.948 | 0.953 | 0.987 | 0.147 | 0.941 | 0.540 | 0.703 | 0.708 | 0.934 | 0.139 | 0.662 | 0.720 | 0.815 | 0.811 | 0.971 | 0.147 | 0.787 |
| Gemini 1.5 Pro | 0.920 | 0.953 | 0.956 | 0.991 | 0.134 | 0.947 | 0.525 | 0.714 | 0.719 | 0.924 | 0.175 | 0.664 | 0.810 | 0.886 | 0.882 | 0.979 | 0.193 | 0.864 |
| Gemini 2.5 Flash-Lite | 0.905 | 0.946 | 0.931 | 0.993 | 0.316 | 0.925 | 0.655 | 0.820 | 0.835 | 0.941 | 0.208 | 0.786 | 0.745 | 0.843 | 0.839 | 0.971 | 0.173 | 0.815 |
| Gemini 2.5 Flash | 0.920 | 0.956 | 0.953 | 0.991 | 0.249 | 0.945 | 0.590 | 0.784 | 0.729 | 0.947 | 0.345 | 0.690 | 0.770 | 0.866 | 0.875 | 0.969 | 0.151 | 0.847 |
| Gemini 2.5 Pro | 0.940 | 0.970 | 0.971 | 0.992 | 0.232 | 0.963 | 0.595 | 0.786 | 0.823 | 0.922 | 0.156 | 0.759 | 0.860 | 0.930 | 0.928 | 0.981 | 0.275 | 0.910 |
| *Open Source VLMs* | | | | | | | | | | | | | | | | | | |
| Gemma3 4B | 0.795 | 0.897 | 0.919 | 0.964 | 0.136 | 0.887 | 0.175 | 0.372 | 0.392 | 0.818 | 0.086 | 0.320 | 0.825 | 0.922 | 0.934 | 0.969 | 0.257 | 0.906 |
| Gemma3 12B | 0.780 | 0.896 | 0.907 | 0.963 | 0.246 | 0.873 | 0.255 | 0.477 | 0.477 | 0.855 | 0.134 | 0.407 | 0.805 | 0.894 | 0.894 | 0.974 | 0.216 | 0.871 |
| Gemma3 27B | 0.860 | 0.924 | 0.904 | 0.987 | 0.324 | 0.893 | 0.230 | 0.416 | 0.420 | 0.860 | 0.094 | 0.362 | 0.890 | 0.949 | 0.954 | 0.983 | 0.245 | 0.937 |
| Deepseek VL2-Tiny | 0.870 | 0.931 | 0.932 | 0.983 | 0.220 | 0.916 | 0.250 | 0.374 | 0.389 | 0.896 | 0.042 | 0.348 | 0.335 | 0.546 | 0.546 | 0.885 | 0.138 | 0.483 |
| Deepseek VL2-Small | 0.885 | 0.932 | 0.913 | 0.992 | 0.307 | 0.906 | 0.555 | 0.724 | 0.750 | 0.927 | 0.113 | 0.696 | 0.645 | 0.761 | 0.764 | 0.958 | 0.122 | 0.732 |
| Deepseek VL2 | 0.840 | 0.925 | 0.929 | 0.975 | 0.272 | 0.906 | 0.560 | 0.755 | 0.780 | 0.921 | 0.169 | 0.718 | 0.695 | 0.771 | 0.772 | 0.974 | 0.085 | 0.752 |
| InternVL2.5 4B-MPO | 0.815 | 0.900 | 0.905 | 0.974 | 0.195 | 0.881 | 0.610 | 0.767 | 0.756 | 0.948 | 0.210 | 0.716 | 0.485 | 0.624 | 0.593 | 0.951 | 0.148 | 0.564 |
| InternVL2.5 8B-MPO | 0.810 | 0.881 | 0.848 | 0.988 | 0.284 | 0.838 | 0.625 | 0.785 | 0.809 | 0.937 | 0.139 | 0.758 | 0.415 | 0.594 | 0.573 | 0.922 | 0.153 | 0.528 |
| InternVL3-1B | 0.755 | 0.869 | 0.893 | 0.959 | 0.118 | 0.856 | 0.405 | 0.532 | 0.514 | 0.942 | 0.099 | 0.484 | 0.450 | 0.600 | 0.602 | 0.930 | 0.101 | 0.560 |
| InternVL3-2B | 0.855 | 0.922 | 0.917 | 0.983 | 0.259 | 0.901 | 0.365 | 0.519 | 0.524 | 0.914 | 0.157 | 0.479 | 0.615 | 0.749 | 0.743 | 0.954 | 0.157 | 0.708 |
| InternVL3-8B | 0.880 | 0.924 | 0.932 | 0.986 | 0.074 | 0.919 | 0.405 | 0.546 | 0.554 | 0.925 | 0.076 | 0.512 | 0.595 | 0.759 | 0.777 | 0.935 | 0.144 | 0.727 |
| InternVL3-9B | 0.830 | 0.916 | 0.927 | 0.973 | 0.199 | 0.902 | 0.415 | 0.613 | 0.624 | 0.903 | 0.131 | 0.563 | 0.485 | 0.656 | 0.646 | 0.931 | 0.156 | 0.601 |
| InternVL3-14B | 0.850 | 0.909 | 0.903 | 0.985 | 0.198 | 0.890 | 0.395 | 0.556 | 0.539 | 0.925 | 0.121 | 0.498 | 0.565 | 0.698 | 0.679 | 0.955 | 0.154 | 0.648 |
| InternVL3-38B | 0.950 | 0.976 | 0.982 | 0.992 | 0.140 | 0.974 | 0.520 | 0.654 | 0.620 | 0.957 | 0.159 | 0.593 | 0.645 | 0.770 | 0.782 | 0.953 | 0.112 | 0.746 |
| InternVL3.5-1B | 0.705 | 0.821 | 0.829 | 0.960 | 0.149 | 0.796 | 0.220 | 0.396 | 0.402 | 0.860 | 0.085 | 0.346 | 0.645 | 0.811 | 0.794 | 0.949 | 0.282 | 0.753 |
| InternVL3.5-2B | 0.750 | 0.845 | 0.848 | 0.970 | 0.148 | 0.823 | 0.140 | 0.273 | 0.286 | 0.837 | 0.047 | 0.239 | 0.780 | 0.879 | 0.896 | 0.966 | 0.129 | 0.865 |
| InternVL3.5-4B | 0.690 | 0.826 | 0.822 | 0.957 | 0.223 | 0.787 | 0.245 | 0.364 | 0.371 | 0.901 | 0.046 | 0.335 | 0.765 | 0.877 | 0.893 | 0.962 | 0.174 | 0.859 |
| InternVL3.5-8B | 0.700 | 0.834 | 0.819 | 0.961 | 0.258 | 0.787 | 0.130 | 0.275 | 0.307 | 0.807 | 0.040 | 0.247 | 0.720 | 0.839 | 0.791 | 0.976 | 0.319 | 0.772 |
| InternVL3.5-14B | 0.720 | 0.836 | 0.841 | 0.962 | 0.171 | 0.809 | 0.140 | 0.304 | 0.303 | 0.825 | 0.078 | 0.250 | 0.815 | 0.909 | 0.924 | 0.969 | 0.177 | 0.895 |
| InternVL3.5-38B | 0.855 | 0.921 | 0.938 | 0.977 | 0.077 | 0.916 | 0.380 | 0.569 | 0.568 | 0.904 | 0.128 | 0.513 | 0.730 | 0.866 | 0.860 | 0.959 | 0.293 | 0.825 |
| InternVL3.5-1B-Thk | 0.705 | 0.834 | 0.846 | 0.955 | 0.164 | 0.809 | 0.245 | 0.421 | 0.413 | 0.877 | 0.100 | 0.363 | 0.630 | 0.818 | 0.809 | 0.939 | 0.303 | 0.760 |
| InternVL3.5-2B-Thk | 0.720 | 0.820 | 0.807 | 0.972 | 0.185 | 0.784 | 0.250 | 0.502 | 0.559 | 0.801 | 0.125 | 0.448 | 0.545 | 0.767 | 0.808 | 0.906 | 0.131 | 0.732 |
| InternVL3.5-4B-Thk | 0.695 | 0.830 | 0.810 | 0.962 | 0.266 | 0.779 | 0.350 | 0.495 | 0.477 | 0.925 | 0.102 | 0.442 | 0.695 | 0.853 | 0.827 | 0.956 | 0.354 | 0.791 |
| InternVL3.5-8B-Thk | 0.700 | 0.839 | 0.826 | 0.959 | 0.268 | 0.792 | 0.240 | 0.436 | 0.459 | 0.850 | 0.084 | 0.391 | 0.715 | 0.834 | 0.814 | 0.968 | 0.246 | 0.788 |
| InternVL3.5-14B-Thk | 0.730 | 0.860 | 0.860 | 0.990 | 0.247 | 0.826 | 0.200 | 0.424 | 0.476 | 0.805 | 0.078 | 0.383 | 0.780 | 0.889 | 0.865 | 0.974 | 0.345 | 0.842 |
| InternVL3.5-38B-Thk | 0.880 | 0.930 | 0.934 | 0.985 | 0.146 | 0.920 | 0.435 | 0.629 | 0.616 | 0.916 | 0.167 | 0.565 | 0.695 | 0.853 | 0.885 | 0.941 | 0.168 | 0.833 |
| VLM-R1-OVD | 0.615 | 0.786 | 0.778 | 0.943 | 0.237 | 0.734 | 0.370 | 0.593 | 0.583 | 0.892 | 0.173 | 0.520 | 0.440 | 0.636 | 0.669 | 0.901 | 0.102 | 0.602 |
| VLM-R1-Math | 0.645 | 0.799 | 0.791 | 0.950 | 0.226 | 0.752 | 0.505 | 0.671 | 0.677 | 0.929 | 0.130 | 0.629 | 0.480 | 0.667 | 0.684 | 0.915 | 0.131 | 0.626 |
| VLM-R1-REC | 0.580 | 0.777 | 0.785 | 0.927 | 0.231 | 0.728 | 0.530 | 0.730 | 0.752 | 0.916 | 0.166 | 0.689 | 0.305 | 0.535 | 0.533 | 0.870 | 0.154 | 0.463 |
| Kimi-VL-A3B-Thk | 0.735 | 0.863 | 0.840 | 0.967 | 0.316 | 0.812 | 0.355 | 0.611 | 0.571 | 0.887 | 0.244 | 0.506 | 0.375 | 0.601 | 0.607 | 0.886 | 0.161 | 0.538 |
| Kimi-VL-A3B-Instruct | 0.850 | 0.926 | 0.908 | 0.983 | 0.365 | 0.858 | 0.625 | 0.761 | 0.764 | 0.951 | 0.147 | 0.727 | 0.630 | 0.746 | 0.745 | 0.959 | 0.124 | 0.715 |
| Kimi-VL-A3B-Thk-2506 | 0.875 | 0.934 | 0.936 | 0.983 | 0.209 | 0.920 | 0.725 | 0.858 | 0.812 | 0.971 | 0.368 | 0.788 | 0.420 | 0.591 | 0.614 | 0.910 | 0.086 | 0.558 |
| LLaVA-Interleave-Qwen-0.5B | 0.710 | 0.797 | 0.792 | 0.973 | 0.128 | 0.771 | 0.205 | 0.334 | 0.345 | 0.878 | 0.047 | 0.303 | 0.460 | 0.604 | 0.584 | 0.942 | 0.128 | 0.550 |
| LLaVA-1.5-7B | 0.810 | 0.890 | 0.878 | 0.980 | 0.244 | 0.860 | 0.280 | 0.529 | 0.545 | 0.846 | 0.148 | 0.461 | 0.320 | 0.545 | 0.541 | 0.877 | 0.154 | 0.474 |
| LLaVA-Next-Mistral-7B | 0.820 | 0.896 | 0.889 | 0.980 | 0.225 | 0.871 | 0.585 | 0.704 | 0.710 | 0.953 | 0.095 | 0.676 | 0.755 | 0.836 | 0.842 | 0.973 | 0.109 | 0.819 |
| LLaVA-Next-Vicuna-7B | 0.720 | 0.829 | 0.819 | 0.968 | 0.198 | 0.793 | 0.340 | 0.481 | 0.496 | 0.910 | 0.059 | 0.452 | 0.520 | 0.701 | 0.711 | 0.925 | 0.151 | 0.658 |
| LLaVA-Interleave-Qwen-7B | 0.895 | 0.946 | 0.923 | 0.992 | 0.399 | 0.916 | 0.420 | 0.594 | 0.620 | 0.907 | 0.082 | 0.563 | 0.570 | 0.696 | 0.692 | 0.952 | 0.119 | 0.660 |
| LLaVA-1.5-13B | 0.760 | 0.873 | 0.874 | 0.966 | 0.227 | 0.844 | 0.450 | 0.620 | 0.659 | 0.909 | 0.061 | 0.599 | 0.620 | 0.776 | 0.774 | 0.946 | 0.195 | 0.732 |
| LLaVA-Next-Vicuna-13B | 0.725 | 0.829 | 0.815 | 0.971 | 0.202 | 0.791 | 0.390 | 0.530 | 0.533 | 0.925 | 0.080 | 0.493 | 0.605 | 0.706 | 0.714 | 0.959 | 0.074 | 0.685 |
| Phi-4 | 0.680 | 0.812 | 0.811 | 0.957 | 0.194 | 0.776 | 0.505 | 0.833 | 0.849 | 0.955 | 0.147 | 0.810 | 0.140 | 0.304 | 0.287 | 0.836 | 0.090 | 0.240 |
| Qwen2.5VL 3B | 0.665 | 0.804 | 0.788 | 0.958 | 0.228 | 0.755 | 0.590 | 0.752 | 0.763 | 0.938 | 0.157 | 0.715 | 0.470 | 0.665 | 0.668 | 0.916 | 0.160 | 0.612 |
| Qwen2.5VL 7B | 0.825 | 0.914 | 0.901 | 0.978 | 0.329 | 0.881 | 0.720 | 0.834 | 0.810 | 0.971 | 0.251 | 0.786 | 0.435 | 0.590 | 0.583 | 0.929 | 0.116 | 0.542 |
| Qwen2.5VL 32B | 0.670 | 0.838 | 0.861 | 0.939 | 0.206 | 0.809 | 0.655 | 0.789 | 0.800 | 0.951 | 0.139 | 0.761 | 0.435 | 0.611 | 0.594 | 0.925 | 0.152 | 0.550 |

Each table contains RE, VE, $\theta$, $r$, $g$, and $A_{adj}$ for every model, with the best values highlighted in green and the worst in red (separately for open-source and proprietary models). Unlike the other three tables, Table 9 (Visual Grounding) presents only the top 12 models by $A_{adj}$ in each group. Many models produced near-zero RE and consequently very low $A_{adj}$ on this task, likely due to the lack of relevant training data, offering little comparative insight. In rare cases, such as for GPT-4o on *Scene Understanding*, a valid solution to the parameter system could not be found.

## E.5  VISUAL GROUNDING

In TDBench, we employ a lenient criteria, centroid containment criterion, for visual grounding evaluation rather than the conventional Intersection over Union (IoU) metric typically used in object detection tasks. The reason is that aerial applications, such as drone navigation scenarios where precise object boundaries are less critical than accurate central positioning as waypoint. Specifically,

Table 7: VLMs in TDBench on Object Localization, Attribute Recognition, Object Counting.

| Model | Object Localization | | | | | | Attribute Recognition | | | | | | Object Counting | | | | | |
|---|---|---|---|---|---|---|---|---|---|---|---|---|---|---|---|---|---|---|
| | RE | VE | $\theta$ | $r$ | $g$ | $A_{adj}$ | RE | VE | $\theta$ | $r$ | $g$ | $A_{adj}$ | RE | VE | $\theta$ | $r$ | $g$ | $A_{adj}$ |
| *Proprietary VLMs* | | | | | | | | | | | | | | | | | | |
| Claude 3.5 Haiku | 0.165 | 0.496 | 0.475 | 0.765 | 0.253 | 0.363 | 0.405 | 0.608 | 0.579 | 0.914 | 0.186 | 0.529 | 0.075 | 0.316 | 0.373 | 0.669 | 0.106 | 0.250 |
| Claude 3.5 Sonnet | 0.335 | 0.627 | 0.541 | 0.884 | 0.326 | 0.478 | 0.525 | 0.714 | 0.711 | 0.927 | 0.189 | 0.659 | 0.135 | 0.394 | 0.397 | 0.763 | 0.151 | 0.303 |
| Claude 3.7 Sonnet | 0.340 | 0.583 | 0.484 | 0.914 | 0.272 | 0.442 | 0.480 | 0.669 | 0.613 | 0.940 | 0.239 | 0.577 | 0.115 | 0.354 | 0.388 | 0.738 | 0.111 | 0.286 |
| Claude 4 Sonnet | 0.420 | 0.693 | 0.664 | 0.890 | 0.302 | 0.591 | 0.490 | 0.704 | 0.737 | 0.903 | 0.146 | 0.665 | 0.100 | 0.399 | 0.427 | 0.695 | 0.178 | 0.297 |
| Claude 4.1 Opus | 0.365 | 0.641 | 0.623 | 0.874 | 0.257 | 0.544 | 0.505 | 0.703 | 0.717 | 0.916 | 0.161 | 0.657 | 0.165 | 0.398 | 0.425 | 0.789 | 0.108 | 0.335 |
| GPT 4o-mini | 0.075 | 0.468 | 0.488 | 0.613 | 0.328 | 0.299 | 0.480 | 0.693 | 0.659 | 0.923 | 0.246 | 0.609 | 0.175 | 0.366 | 0.303 | 0.871 | 0.146 | 0.264 |
| GPT-4o | 0.435 | 0.728 | 0.722 | 0.879 | 0.334 | 0.635 | 0.610 | 0.796 | 0.800 | 0.934 | 0.245 | 0.747 | 0.200 | 0.465 | 0.370 | 0.855 | 0.236 | 0.316 |
| GPT-4.1 Nano | 0.570 | 0.811 | 0.874 | 0.899 | 0.207 | 0.785 | 0.700 | 0.839 | 0.865 | 0.949 | 0.137 | 0.820 | 0.185 | 0.453 | 0.411 | 0.818 | 0.197 | 0.336 |
| GPT-4.1 | 0.660 | 0.839 | 0.846 | 0.940 | 0.287 | 0.795 | 0.680 | 0.818 | 0.779 | 0.966 | 0.293 | 0.753 | 0.235 | 0.477 | 0.437 | 0.856 | 0.184 | 0.374 |
| OpenAI o3 | 0.780 | 0.891 | 0.900 | 0.965 | 0.232 | 0.868 | 0.720 | 0.838 | 0.812 | 0.970 | 0.264 | 0.788 | 0.215 | 0.480 | 0.421 | 0.844 | 0.215 | 0.356 |
| GPT-5 mini | 0.575 | 0.826 | 0.831 | 0.910 | 0.414 | 0.756 | 0.700 | 0.821 | 0.827 | 0.959 | 0.163 | 0.793 | 0.215 | 0.484 | 0.449 | 0.831 | 0.201 | 0.373 |
| GPT-5 | 0.770 | 0.887 | 0.886 | 0.965 | 0.284 | 0.855 | 0.700 | 0.838 | 0.841 | 0.955 | 0.217 | 0.803 | 0.190 | 0.465 | 0.436 | 0.812 | 0.197 | 0.354 |
| Gemini 1.5 Flash | 0.600 | 0.841 | 0.926 | 0.897 | 0.144 | 0.831 | 0.780 | 0.869 | 0.861 | 0.975 | 0.207 | 0.840 | 0.255 | 0.494 | 0.506 | 0.842 | 0.136 | 0.426 |
| Gemini 1.5 Pro | 0.715 | 0.892 | 0.928 | 0.937 | 0.325 | 0.869 | 0.740 | 0.860 | 0.851 | 0.966 | 0.259 | 0.821 | 0.285 | 0.492 | 0.448 | 0.893 | 0.168 | 0.400 |
| Gemini 2.5 Flash-Lite | 0.460 | 0.721 | 0.744 | 0.886 | 0.241 | 0.660 | 0.695 | 0.819 | 0.792 | 0.968 | 0.252 | 0.766 | 0.125 | 0.374 | 0.376 | 0.759 | 0.142 | 0.285 |
| Gemini 2.5 Flash | 0.585 | 0.795 | 0.685 | 0.956 | 0.445 | 0.655 | 0.755 | 0.856 | 0.834 | 0.975 | 0.259 | 0.813 | 0.145 | 0.432 | 0.422 | 0.765 | 0.190 | 0.323 |
| Gemini 2.5 Pro | 0.780 | 0.901 | 0.900 | 0.964 | 0.331 | 0.868 | 0.805 | 0.900 | 0.913 | 0.969 | 0.177 | 0.885 | 0.210 | 0.499 | 0.483 | 0.811 | 0.207 | 0.392 |
| *Open Source VLMs* | | | | | | | | | | | | | | | | | | |
| Gemma3 4B | 0.035 | 0.400 | 0.074 | 0.684 | 0.377 | 0.051 | 0.435 | 0.677 | 0.698 | 0.888 | 0.190 | 0.620 | 0.035 | 0.301 | 0.281 | 0.590 | 0.188 | 0.166 |
| Gemma3 12B | 0.280 | 0.593 | 0.453 | 0.880 | 0.355 | 0.399 | 0.290 | 0.666 | 0.867 | 0.761 | 0.053 | 0.659 | 0.130 | 0.414 | 0.438 | 0.738 | 0.161 | 0.323 |
| Gemma3 27B | 0.440 | 0.689 | 0.646 | 0.907 | 0.290 | 0.586 | 0.590 | 0.770 | 0.757 | 0.939 | 0.242 | 0.711 | 0.125 | 0.365 | 0.363 | 0.766 | 0.137 | 0.278 |
| Deepseek VL2-Tiny | 0.130 | 0.410 | 0.395 | 0.756 | 0.184 | 0.299 | 0.610 | 0.774 | 0.789 | 0.938 | 0.160 | 0.740 | 0.165 | 0.369 | 0.319 | 0.848 | 0.145 | 0.270 |
| Deepseek VL2-Small | 0.375 | 0.705 | 0.697 | 0.853 | 0.364 | 0.595 | 0.725 | 0.831 | 0.835 | 0.965 | 0.153 | 0.806 | 0.235 | 0.395 | 0.361 | 0.898 | 0.110 | 0.325 |
| Deepseek VL2 | 0.365 | 0.723 | 0.820 | 0.816 | 0.297 | 0.669 | 0.680 | 0.830 | 0.857 | 0.944 | 0.150 | 0.809 | 0.235 | 0.398 | 0.358 | 0.900 | 0.117 | 0.322 |
| InternVL2.5 4B-MPO | 0.180 | 0.531 | 0.363 | 0.826 | 0.363 | 0.300 | 0.570 | 0.754 | 0.741 | 0.936 | 0.233 | 0.693 | 0.260 | 0.432 | 0.415 | 0.890 | 0.108 | 0.369 |
| InternVL2.5 8B-MPO | 0.390 | 0.649 | 0.616 | 0.891 | 0.261 | 0.549 | 0.630 | 0.784 | 0.781 | 0.948 | 0.190 | 0.740 | 0.230 | 0.434 | 0.443 | 0.849 | 0.103 | 0.376 |
| InternVL3-1B | 0.110 | 0.459 | 0.441 | 0.702 | 0.267 | 0.309 | 0.500 | 0.699 | 0.696 | 0.921 | 0.192 | 0.640 | 0.300 | 0.427 | 0.446 | 0.906 | 0.043 | 0.404 |
| InternVL3-2B | 0.270 | 0.534 | 0.502 | 0.856 | 0.209 | 0.430 | 0.595 | 0.761 | 0.741 | 0.946 | 0.233 | 0.701 | 0.285 | 0.435 | 0.448 | 0.893 | 0.063 | 0.400 |
| InternVL3-8B | 0.570 | 0.769 | 0.724 | 0.941 | 0.317 | 0.681 | 0.660 | 0.807 | 0.820 | 0.947 | 0.171 | 0.777 | 0.165 | 0.414 | 0.454 | 0.776 | 0.113 | 0.352 |
| InternVL3-9B | 0.640 | 0.815 | 0.767 | 0.954 | 0.356 | 0.732 | 0.630 | 0.794 | 0.789 | 0.945 | 0.228 | 0.746 | 0.315 | 0.465 | 0.454 | 0.913 | 0.093 | 0.414 |
| InternVL3-14B | 0.595 | 0.823 | 0.883 | 0.906 | 0.191 | 0.800 | 0.650 | 0.792 | 0.783 | 0.954 | 0.209 | 0.747 | 0.320 | 0.490 | 0.496 | 0.896 | 0.091 | 0.444 |
| InternVL3-38B | 0.795 | 0.911 | 0.907 | 0.967 | 0.366 | 0.877 | 0.800 | 0.874 | 0.860 | 0.982 | 0.208 | 0.845 | 0.340 | 0.475 | 0.455 | 0.930 | 0.095 | 0.423 |
| InternVL3.5-1B | 0.330 | 0.588 | 0.566 | 0.873 | 0.215 | 0.494 | 0.540 | 0.703 | 0.729 | 0.928 | 0.098 | 0.676 | 0.360 | 0.482 | 0.479 | 0.931 | 0.070 | 0.446 |
| InternVL3.5-2B | 0.410 | 0.679 | 0.567 | 0.918 | 0.366 | 0.520 | 0.555 | 0.730 | 0.715 | 0.925 | 0.240 | 0.662 | 0.315 | 0.474 | 0.474 | 0.903 | 0.133 | 0.428 |
| InternVL3.5-4B | 0.625 | 0.815 | 0.805 | 0.938 | 0.307 | 0.755 | 0.555 | 0.720 | 0.716 | 0.938 | 0.170 | 0.672 | 0.280 | 0.487 | 0.478 | 0.875 | 0.133 | 0.418 |
| InternVL3.5-8B | 0.730 | 0.875 | 0.881 | 0.954 | 0.291 | 0.840 | 0.540 | 0.733 | 0.734 | 0.926 | 0.198 | 0.680 | 0.320 | 0.504 | 0.478 | 0.904 | 0.137 | 0.432 |
| InternVL3.5-14B | 0.710 | 0.874 | 0.899 | 0.943 | 0.262 | 0.847 | 0.515 | 0.709 | 0.697 | 0.927 | 0.207 | 0.646 | 0.300 | 0.499 | 0.523 | 0.870 | 0.142 | 0.455 |
| InternVL3.5-38B | 0.820 | 0.919 | 0.935 | 0.968 | 0.211 | 0.905 | 0.495 | 0.713 | 0.724 | 0.909 | 0.197 | 0.658 | 0.355 | 0.564 | 0.533 | 0.903 | 0.176 | 0.481 |
| InternVL3.5-1B-Thk | 0.210 | 0.459 | 0.447 | 0.827 | 0.160 | 0.370 | 0.525 | 0.693 | 0.704 | 0.929 | 0.130 | 0.654 | 0.185 | 0.429 | 0.379 | 0.835 | 0.181 | 0.316 |
| InternVL3.5-2B-Thk | 0.245 | 0.532 | 0.405 | 0.878 | 0.297 | 0.356 | 0.535 | 0.679 | 0.711 | 0.884 | 0.114 | 0.628 | 0.185 | 0.444 | 0.450 | 0.801 | 0.152 | 0.360 |
| InternVL3.5-4B-Thk | 0.525 | 0.781 | 0.814 | 0.896 | 0.280 | 0.729 | 0.555 | 0.734 | 0.723 | 0.936 | 0.206 | 0.677 | 0.175 | 0.451 | 0.477 | 0.778 | 0.153 | 0.371 |
| InternVL3.5-8B-Thk | 0.670 | 0.853 | 0.861 | 0.939 | 0.319 | 0.808 | 0.570 | 0.761 | 0.789 | 0.922 | 0.160 | 0.728 | 0.260 | 0.510 | 0.529 | 0.837 | 0.142 | 0.443 |
| InternVL3.5-14B-Thk | 0.650 | 0.853 | 0.805 | 0.944 | 0.474 | 0.760 | 0.570 | 0.748 | 0.760 | 0.931 | 0.168 | 0.707 | 0.325 | 0.539 | 0.513 | 0.892 | 0.167 | 0.458 |
| InternVL3.5-38B-Thk | 0.790 | 0.901 | 0.914 | 0.964 | 0.231 | 0.882 | 0.555 | 0.761 | 0.742 | 0.929 | 0.279 | 0.689 | 0.325 | 0.545 | 0.527 | 0.886 | 0.165 | 0.467 |
| VLM-R1-OVD | 0.445 | 0.731 | 0.698 | 0.891 | 0.363 | 0.622 | 0.525 | 0.738 | 0.742 | 0.917 | 0.221 | 0.681 | 0.160 | 0.407 | 0.416 | 0.787 | 0.137 | 0.328 |
| VLM-R1-Math | 0.495 | 0.772 | 0.668 | 0.920 | 0.476 | 0.614 | 0.575 | 0.764 | 0.759 | 0.937 | 0.210 | 0.711 | 0.145 | 0.367 | 0.347 | 0.804 | 0.136 | 0.279 |
| VLM-R1-REC | 0.330 | 0.641 | 0.561 | 0.871 | 0.347 | 0.489 | 0.455 | 0.711 | 0.752 | 0.882 | 0.193 | 0.663 | 0.120 | 0.354 | 0.298 | 0.796 | 0.166 | 0.237 |
| Kimi-VL-A3B-Thk | 0.160 | 0.455 | 0.330 | 0.830 | 0.271 | 0.274 | 0.555 | 0.749 | 0.749 | 0.928 | 0.216 | 0.694 | 0.060 | 0.368 | 0.423 | 0.612 | 0.188 | 0.259 |
| Kimi-VL-A3B-Instruct | 0.555 | 0.776 | 0.800 | 0.912 | 0.231 | 0.730 | 0.710 | 0.825 | 0.835 | 0.960 | 0.161 | 0.802 | 0.260 | 0.441 | 0.431 | 0.881 | 0.093 | 0.380 |
| Kimi-VL-A3B-Thk-2506 | 0.510 | 0.746 | 0.686 | 0.926 | 0.353 | 0.635 | 0.650 | 0.806 | 0.793 | 0.951 | 0.252 | 0.754 | 0.120 | 0.331 | 0.327 | 0.778 | 0.114 | 0.254 |
| LLaVA-Interleave-Qwen-0.5B | 0.015 | 0.270 | 0.015 | 0.917 | 0.260 | 0.014 | 0.420 | 0.630 | 0.629 | 0.904 | 0.165 | 0.569 | 0.060 | 0.217 | 0.184 | 0.756 | 0.096 | 0.139 |
| LLaVA-1.5-7B | 0.115 | 0.535 | 0.733 | 0.627 | 0.282 | 0.460 | 0.385 | 0.644 | 0.638 | 0.881 | 0.222 | 0.562 | 0.180 | 0.361 | 0.359 | 0.842 | 0.093 | 0.302 |
| LLaVA-Next-Mistral-7B | 0.700 | 0.853 | 0.853 | 0.951 | 0.278 | 0.812 | 0.690 | 0.820 | 0.837 | 0.953 | 0.138 | 0.797 | 0.110 | 0.314 | 0.277 | 0.793 | 0.130 | 0.220 |
| LLaVA-Next-Vicuna-7B | 0.435 | 0.704 | 0.609 | 0.915 | 0.374 | 0.558 | 0.575 | 0.750 | 0.749 | 0.936 | 0.196 | 0.701 | 0.105 | 0.314 | 0.295 | 0.772 | 0.122 | 0.228 |
| LLaVA-Interleave-Qwen-7B | 0.170 | 0.514 | 0.472 | 0.771 | 0.284 | 0.364 | 0.600 | 0.807 | 0.813 | 0.949 | 0.191 | 0.772 | 0.175 | 0.351 | 0.322 | 0.858 | 0.110 | 0.277 |
| LLaVA-1.5-13B | 0.385 | 0.718 | 0.778 | 0.838 | 0.297 | 0.652 | 0.535 | 0.733 | 0.747 | 0.920 | 0.179 | 0.687 | 0.205 | 0.345 | 0.326 | 0.891 | 0.082 | 0.290 |
| LLaVA-Next-Vicuna-13B | 0.470 | 0.741 | 0.706 | 0.901 | 0.358 | 0.636 | 0.660 | 0.789 | 0.801 | 0.953 | 0.128 | 0.763 | 0.085 | 0.278 | 0.212 | 0.795 | 0.138 | 0.169 |
| Phi-4 | 0.010 | 0.244 | - | - | - | - | 0.385 | 0.600 | 0.598 | 0.896 | 0.160 | 0.536 | 0.050 | 0.224 | 0.225 | 0.686 | 0.093 | 0.154 |
| Qwen2.5VL 3B | 0.470 | 0.726 | 0.709 | 0.901 | 0.300 | 0.639 | 0.595 | 0.762 | 0.762 | 0.940 | 0.195 | 0.716 | 0.185 | 0.401 | 0.391 | 0.829 | 0.126 | 0.324 |
| Qwen2.5VL 7B | 0.715 | 0.863 | 0.862 | 0.954 | 0.290 | 0.823 | 0.665 | 0.825 | 0.780 | 0.960 | 0.347 | 0.749 | 0.125 | 0.326 | 0.311 | 0.796 | 0.114 | 0.248 |
| Qwen2.5VL 32B | 0.600 | 0.824 | 0.757 | 0.939 | 0.464 | 0.711 | 0.630 | 0.814 | 0.840 | 0.931 | 0.202 | 0.781 | 0.115 | 0.321 | 0.282 | 0.799 | 0.134 | 0.225 |

a prediction is considered successful if the predicted object's centroid falls within the ground truth bounding box, enabling effective target localization for hovering operations. While boundary precision is less relevant in many aerial contexts, we nevertheless present comparative performance analysis using both centroid containment and IoU thresholds in Table 10. Note that value of IoU here is obtained by the calculating the mean in 4 rotations dataset, whereas centroid performance is obtained under RE. We also show some examples of grounding results from some models in Figure 11 for reference.

Table 8: VLMs in TDBench on Attribute Comparison, Dynamic Temporal, Spatial Relationship.

| Model | Attribute Comparison | | | | | | Dynamic Temporal | | | | | | Spatial Relationship | | | | | |
|---|---|---|---|---|---|---|---|---|---|---|---|---|---|---|---|---|---|---|
| | RE | VE | $\theta$ | $r$ | $g$ | $A_{adj}$ | RE | VE | $\theta$ | $r$ | $g$ | $A_{adj}$ | RE | VE | $\theta$ | $r$ | $g$ | $A_{adj}$ |
| *Proprietary VLMs* | | | | | | | | | | | | | | | | | | |
| Claude 3.5 Haiku | 0.615 | 0.670 | 0.669 | 0.979 | 0.045 | 0.655 | 0.200 | 0.375 | 0.333 | 0.880 | 0.123 | 0.293 | 0.190 | 0.501 | 0.576 | 0.758 | 0.153 | 0.436 |
| Claude 3.5 Sonnet | 0.510 | 0.669 | 0.676 | 0.932 | 0.119 | 0.630 | 0.295 | 0.610 | 0.501 | 0.871 | 0.349 | 0.436 | 0.410 | 0.715 | 0.539 | 0.920 | 0.475 | 0.496 |
| Claude 3.7 Sonnet | 0.610 | 0.693 | 0.696 | 0.967 | 0.062 | 0.674 | 0.245 | 0.619 | 0.505 | 0.822 | 0.411 | 0.415 | 0.475 | 0.723 | 0.700 | 0.907 | 0.293 | 0.635 |
| Claude 4 Sonnet | 0.545 | 0.679 | 0.708 | 0.937 | 0.053 | 0.663 | 0.200 | 0.519 | 0.468 | 0.806 | 0.266 | 0.377 | 0.480 | 0.759 | 0.782 | 0.884 | 0.308 | 0.692 |
| Claude 4.1 Opus | 0.530 | 0.679 | 0.687 | 0.937 | 0.112 | 0.644 | 0.210 | 0.530 | 0.471 | 0.814 | 0.277 | 0.383 | 0.485 | 0.776 | 0.867 | 0.865 | 0.199 | 0.750 |
| GPT 4o-mini | 0.420 | 0.630 | 0.660 | 0.893 | 0.119 | 0.590 | 0.145 | 0.432 | 0.383 | 0.783 | 0.215 | 0.300 | 0.025 | 0.393 | 0.983 | 0.399 | 0.004 | 0.392 |
| GPT-4o | 0.420 | 0.647 | 0.634 | 0.902 | 0.207 | 0.572 | 0.225 | 0.529 | 0.444 | 0.840 | 0.280 | 0.373 | 0.415 | 0.720 | 0.695 | 0.876 | 0.364 | 0.609 |
| GPT-4.1 Nano | 0.600 | 0.703 | 0.708 | 0.959 | 0.079 | 0.679 | 0.310 | 0.627 | 0.456 | 0.898 | 0.401 | 0.409 | 0.665 | 0.860 | 0.863 | 0.936 | 0.382 | 0.808 |
| GPT-4.1 | 0.525 | 0.677 | 0.683 | 0.936 | 0.120 | 0.639 | 0.240 | 0.536 | 0.483 | 0.838 | 0.254 | 0.405 | 0.600 | 0.815 | 0.691 | 0.958 | 0.495 | 0.662 |
| OpenAI o3 | 0.550 | 0.714 | 0.694 | 0.937 | 0.192 | 0.655 | 0.280 | 0.605 | 0.409 | 0.897 | 0.403 | 0.367 | 0.770 | 0.909 | 0.903 | 0.960 | 0.434 | 0.867 |
| GPT-5 mini | 0.540 | 0.705 | 0.710 | 0.934 | 0.145 | 0.663 | 0.210 | 0.481 | 0.413 | 0.843 | 0.227 | 0.348 | 0.855 | 0.934 | 0.951 | 0.974 | 0.153 | 0.926 |
| GPT-5 | 0.580 | 0.733 | 0.731 | 0.944 | 0.158 | 0.690 | 0.340 | 0.679 | 0.324 | 0.966 | 0.541 | 0.313 | 0.805 | 0.922 | 0.958 | 0.957 | 0.119 | 0.918 |
| Gemini 1.5 Flash | 0.475 | 0.649 | 0.671 | 0.917 | 0.101 | 0.616 | 0.190 | 0.489 | 0.440 | 0.809 | 0.222 | 0.356 | 0.635 | 0.829 | 0.819 | 0.938 | 0.337 | 0.768 |
| Gemini 1.5 Pro | 0.565 | 0.704 | 0.696 | 0.949 | 0.142 | 0.661 | 0.185 | 0.469 | 0.485 | 0.785 | 0.170 | 0.381 | 0.620 | 0.845 | 0.857 | 0.921 | 0.388 | 0.790 |
| Gemini 2.5 Flash-Lite | 0.410 | 0.639 | 0.656 | 0.889 | 0.162 | 0.583 | 0.170 | 0.505 | 0.455 | 0.778 | 0.277 | 0.354 | 0.620 | 0.846 | 0.865 | 0.919 | 0.380 | 0.795 |
| Gemini 2.5 Flash | 0.455 | 0.691 | 0.663 | 0.909 | 0.262 | 0.603 | 0.225 | 0.525 | 0.515 | 0.812 | 0.220 | 0.418 | 0.820 | 0.921 | 0.943 | 0.966 | 0.186 | 0.911 |
| Gemini 2.5 Pro | 0.575 | 0.744 | 0.743 | 0.938 | 0.182 | 0.697 | 0.240 | 0.535 | 0.542 | 0.815 | 0.203 | 0.442 | 0.825 | 0.925 | 0.943 | 0.967 | 0.231 | 0.912 |
| *Open Source VLMs* | | | | | | | | | | | | | | | | | | |
| Gemma3 4B | 0.260 | 0.550 | 0.523 | 0.838 | 0.233 | 0.439 | 0.140 | 0.429 | 0.463 | 0.741 | 0.160 | 0.343 | 0.020 | 0.331 | 0.567 | 0.429 | 0.203 | 0.243 |
| Gemma3 12B | 0.450 | 0.634 | 0.610 | 0.927 | 0.176 | 0.565 | 0.165 | 0.415 | 0.472 | 0.769 | 0.098 | 0.363 | 0.135 | 0.550 | 0.816 | 0.638 | 0.162 | 0.520 |
| Gemma3 27B | 0.490 | 0.676 | 0.670 | 0.925 | 0.172 | 0.619 | 0.145 | 0.386 | 0.422 | 0.766 | 0.109 | 0.323 | 0.495 | 0.744 | 0.673 | 0.923 | 0.375 | 0.621 |
| Deepseek VL2-Tiny | 0.330 | 0.603 | 0.607 | 0.858 | 0.208 | 0.521 | 0.085 | 0.350 | 0.284 | 0.737 | 0.196 | 0.209 | 0.085 | 0.463 | 0.675 | 0.595 | 0.187 | 0.402 |
| Deepseek VL2-Small | 0.560 | 0.666 | 0.678 | 0.953 | 0.061 | 0.647 | 0.150 | 0.471 | 0.474 | 0.749 | 0.222 | 0.354 | 0.325 | 0.650 | 0.726 | 0.818 | 0.205 | 0.594 |
| Deepseek VL2 | 0.595 | 0.669 | 0.672 | 0.970 | 0.052 | 0.652 | 0.155 | 0.444 | 0.389 | 0.793 | 0.222 | 0.308 | 0.345 | 0.665 | 0.593 | 0.868 | 0.368 | 0.515 |
| InternVL2.5 4B-MPO | 0.415 | 0.637 | 0.638 | 0.898 | 0.178 | 0.573 | 0.145 | 0.405 | 0.375 | 0.788 | 0.175 | 0.296 | 0.140 | 0.566 | 0.871 | 0.633 | 0.115 | 0.551 |
| InternVL2.5 8B-MPO | 0.455 | 0.593 | 0.596 | 0.935 | 0.088 | 0.557 | 0.150 | 0.497 | 0.500 | 0.740 | 0.206 | 0.395 | 0.440 | 0.740 | 0.662 | 0.897 | 0.433 | 0.594 |
| InternVL3-1B | 0.370 | 0.546 | 0.533 | 0.913 | 0.128 | 0.486 | 0.120 | 0.360 | 0.423 | 0.730 | 0.089 | 0.309 | 0.010 | 0.338 | - | - | - | - |
| InternVL3-2B | 0.420 | 0.645 | 0.665 | 0.891 | 0.156 | 0.593 | 0.135 | 0.380 | 0.419 | 0.753 | 0.111 | 0.316 | 0.140 | 0.534 | 0.255 | 0.815 | 0.437 | 0.208 |
| InternVL3-8B | 0.420 | 0.584 | 0.579 | 0.923 | 0.118 | 0.534 | 0.160 | 0.411 | 0.440 | 0.777 | 0.125 | 0.341 | 0.505 | 0.772 | 0.697 | 0.917 | 0.318 | 0.639 |
| InternVL3-9B | 0.450 | 0.649 | 0.669 | 0.906 | 0.130 | 0.606 | 0.145 | 0.401 | 0.476 | 0.743 | 0.091 | 0.354 | 0.595 | 0.807 | 0.813 | 0.924 | 0.300 | 0.751 |
| InternVL3-14B | 0.525 | 0.652 | 0.654 | 0.947 | 0.097 | 0.619 | 0.150 | 0.436 | 0.439 | 0.764 | 0.180 | 0.335 | 0.585 | 0.820 | 0.805 | 0.921 | 0.402 | 0.742 |
| InternVL3-38B | 0.635 | 0.728 | 0.733 | 0.965 | 0.075 | 0.707 | 0.190 | 0.453 | 0.450 | 0.806 | 0.163 | 0.363 | 0.720 | 0.875 | 0.821 | 0.965 | 0.469 | 0.793 |
| InternVL3.5-1B | 0.335 | 0.516 | 0.517 | 0.897 | 0.109 | 0.464 | 0.075 | 0.352 | 0.482 | 0.628 | 0.096 | 0.303 | 0.150 | 0.497 | 0.562 | 0.718 | 0.215 | 0.404 |
| InternVL3.5-2B | 0.295 | 0.540 | 0.542 | 0.859 | 0.163 | 0.465 | 0.155 | 0.393 | 0.452 | 0.765 | 0.085 | 0.346 | 0.365 | 0.642 | 0.653 | 0.864 | 0.226 | 0.564 |
| InternVL3.5-4B | 0.440 | 0.556 | 0.544 | 0.948 | 0.088 | 0.516 | 0.165 | 0.443 | 0.513 | 0.753 | 0.116 | 0.386 | 0.435 | 0.714 | 0.751 | 0.872 | 0.254 | 0.655 |
| InternVL3.5-8B | 0.395 | 0.554 | 0.531 | 0.928 | 0.129 | 0.493 | 0.135 | 0.398 | 0.440 | 0.744 | 0.125 | 0.328 | 0.495 | 0.752 | 0.721 | 0.908 | 0.349 | 0.655 |
| InternVL3.5-14B | 0.495 | 0.615 | 0.592 | 0.956 | 0.120 | 0.566 | 0.150 | 0.421 | 0.474 | 0.750 | 0.125 | 0.355 | 0.570 | 0.801 | 0.846 | 0.906 | 0.228 | 0.766 |
| InternVL3.5-38B | 0.440 | 0.621 | 0.607 | 0.922 | 0.155 | 0.560 | 0.205 | 0.440 | 0.438 | 0.827 | 0.138 | 0.362 | 0.660 | 0.856 | 0.811 | 0.947 | 0.469 | 0.767 |
| InternVL3.5-1B-Thk | 0.365 | 0.596 | 0.541 | 0.905 | 0.232 | 0.490 | 0.060 | 0.347 | 0.529 | 0.580 | 0.086 | 0.307 | 0.055 | 0.417 | 0.666 | 0.535 | 0.183 | 0.356 |
| InternVL3.5-2B-Thk | 0.255 | 0.586 | 0.571 | 0.815 | 0.282 | 0.465 | 0.065 | 0.345 | 0.463 | 0.612 | 0.115 | 0.283 | 0.225 | 0.591 | 0.515 | 0.805 | 0.364 | 0.414 |
| InternVL3.5-4B-Thk | 0.345 | 0.551 | 0.551 | 0.889 | 0.136 | 0.490 | 0.115 | 0.443 | 0.487 | 0.751 | 0.150 | 0.366 | 0.340 | 0.671 | 0.650 | 0.847 | 0.345 | 0.550 |
| InternVL3.5-8B-Thk | 0.320 | 0.578 | 0.529 | 0.881 | 0.237 | 0.466 | 0.115 | 0.399 | 0.472 | 0.702 | 0.127 | 0.332 | 0.435 | 0.725 | 0.681 | 0.891 | 0.371 | 0.606 |
| InternVL3.5-14B-Thk | 0.440 | 0.635 | 0.624 | 0.916 | 0.168 | 0.572 | 0.145 | 0.420 | 0.490 | 0.738 | 0.115 | 0.361 | 0.520 | 0.789 | 0.842 | 0.886 | 0.270 | 0.746 |
| InternVL3.5-38B-Thk | 0.410 | 0.629 | 0.622 | 0.901 | 0.181 | 0.560 | 0.215 | 0.460 | 0.452 | 0.830 | 0.155 | 0.375 | 0.640 | 0.850 | 0.874 | 0.924 | 0.332 | 0.808 |
| VLM-R1-OVD | 0.200 | 0.509 | 0.544 | 0.778 | 0.188 | 0.423 | 0.120 | 0.420 | 0.535 | 0.688 | 0.111 | 0.368 | 0.270 | 0.621 | 0.620 | 0.810 | 0.314 | 0.502 |
| VLM-R1-Math | 0.335 | 0.556 | 0.560 | 0.879 | 0.145 | 0.492 | 0.145 | 0.426 | 0.518 | 0.727 | 0.103 | 0.376 | 0.320 | 0.649 | 0.706 | 0.820 | 0.238 | 0.579 |
| VLM-R1-REC | 0.390 | 0.611 | 0.641 | 0.883 | 0.125 | 0.566 | 0.105 | 0.404 | 0.538 | 0.665 | 0.100 | 0.358 | 0.205 | 0.593 | 0.583 | 0.764 | 0.353 | 0.446 |
| Kimi-VL-A3B-Thk | 0.305 | 0.633 | 0.537 | 0.862 | 0.366 | 0.463 | 0.120 | 0.400 | 0.318 | 0.781 | 0.223 | 0.248 | 0.245 | 0.621 | 0.318 | 0.896 | 0.493 | 0.285 |
| Kimi-VL-A3B-Instruct | 0.370 | 0.545 | 0.540 | 0.910 | 0.117 | 0.491 | 0.120 | 0.379 | 0.355 | 0.762 | 0.168 | 0.270 | 0.425 | 0.682 | 0.703 | 0.881 | 0.211 | 0.620 |
| Kimi-VL-A3B-Thk-2506 | 0.390 | 0.576 | 0.595 | 0.900 | 0.100 | 0.536 | 0.180 | 0.484 | 0.313 | 0.863 | 0.375 | 0.270 | 0.640 | 0.719 | 0.640 | 0.912 | 0.375 | 0.584 |
| LLaVA-Interleave-Qwen-0.5B | 0.660 | 0.672 | 0.675 | 0.994 | 0.004 | 0.671 | 0.135 | 0.312 | 0.278 | 0.835 | 0.112 | 0.232 | 0.005 | 0.233 | 0.030 | 0.545 | 0.223 | 0.016 |
| LLaVA-1.5-7B | 0.635 | 0.662 | 0.659 | 0.991 | 0.027 | 0.653 | 0.090 | 0.356 | 0.400 | 0.688 | 0.135 | 0.275 | 0.040 | 0.366 | - | - | - | - |
| LLaVA-Next-Mistral-7B | 0.560 | 0.670 | 0.668 | 0.957 | 0.093 | 0.639 | 0.125 | 0.362 | 0.428 | 0.735 | 0.084 | 0.314 | 0.175 | 0.566 | 0.519 | 0.753 | 0.365 | 0.390 |
| LLaVA-Next-Vicuna-7B | 0.655 | 0.675 | 0.669 | 0.995 | 0.028 | 0.666 | 0.145 | 0.372 | 0.415 | 0.769 | 0.091 | 0.319 | 0.080 | 0.482 | 0.095 | 0.825 | 0.447 | 0.078 |
| LLaVA-Interleave-Qwen-7B | 0.490 | 0.647 | 0.615 | 0.945 | 0.173 | 0.581 | 0.155 | 0.394 | 0.437 | 0.772 | 0.100 | 0.337 | 0.020 | 0.360 | 0.719 | 0.403 | 0.251 | 0.290 |
| LLaVA-1.5-13B | 0.495 | 0.620 | 0.605 | 0.951 | 0.112 | 0.576 | 0.120 | 0.393 | 0.461 | 0.714 | 0.118 | 0.329 | 0.090 | 0.458 | 0.563 | 0.630 | 0.235 | 0.355 |
| LLaVA-Next-Vicuna-13B | 0.660 | 0.679 | 0.680 | 0.993 | 0.012 | 0.675 | 0.145 | 0.390 | 0.430 | 0.762 | 0.109 | 0.328 | 0.160 | 0.544 | 0.664 | 0.699 | 0.236 | 0.465 |
| Phi-4 | 0.520 | 0.621 | 0.616 | 0.958 | 0.080 | 0.591 | 0.105 | 0.393 | 0.433 | 0.701 | 0.157 | 0.303 | 0.005 | 0.234 | - | - | - | - |
| Qwen2.5VL 3B | 0.265 | 0.516 | 0.459 | 0.871 | 0.216 | 0.400 | 0.130 | 0.415 | 0.494 | 0.716 | 0.121 | 0.354 | 0.250 | 0.603 | 0.627 | 0.793 | 0.282 | 0.497 |
| Qwen2.5VL 7B | 0.465 | 0.621 | 0.623 | 0.929 | 0.111 | 0.579 | 0.165 | 0.424 | 0.472 | 0.769 | 0.115 | 0.363 | 0.580 | 0.819 | 0.835 | 0.912 | 0.348 | 0.761 |
| Qwen2.5VL 32B | 0.375 | 0.589 | 0.617 | 0.883 | 0.115 | 0.545 | 0.205 | 0.459 | 0.487 | 0.805 | 0.130 | 0.392 | 0.650 | 0.851 | 0.865 | 0.930 | 0.344 | 0.805 |

Table 9: Top 12 proprietary and open-source VLMs on Visual Grounding in TDBench. Only the best value in each group is highlighted; unlisted models show substantially lower RE and $A_{adj}$.

| Proprietary VLMs | | | | | | | Open-Source VLMs | | | | | | |
|---|---|---|---|---|---|---|---|---|---|---|---|---|---|
| Model | RE | VE | $\theta$ | $r$ | $g$ | $A_{adj}$ | Model | RE | VE | $\theta$ | $r$ | $g$ | $A_{adj}$ |
| Gemini 2.5 Pro | 0.280 | 0.716 | 0.979 | 0.731 | 0.012 | 0.716 | LLaVA-1.5-13B | 0.610 | 0.829 | 0.836 | 0.923 | 0.346 | 0.772 |
| Gemini 1.5 Pro | 0.360 | 0.745 | 0.928 | 0.789 | 0.177 | 0.732 | LLaVA-1.5-7B | 0.370 | 0.664 | 0.690 | 0.855 | 0.238 | 0.590 |
| Gemini 2.5 Flash | 0.330 | 0.650 | 0.561 | 0.870 | 0.368 | 0.488 | Qwen2.5VL 32B | 0.405 | 0.589 | 0.580 | 0.914 | 0.140 | 0.530 |
| GPT-4.1 | 0.215 | 0.605 | 0.688 | 0.746 | 0.295 | 0.513 | LLaVA-Next-Vicuna-13B | 0.285 | 0.590 | 0.571 | 0.839 | 0.259 | 0.479 |
| Gemini 1.5 Flash | 0.220 | 0.450 | 0.483 | 0.822 | 0.103 | 0.397 | LLaVA-Next-Mistral-7B | 0.305 | 0.573 | 0.446 | 0.906 | 0.304 | 0.404 |
| Gemini 2.5 Flash-Lite | 0.210 | 0.591 | 0.492 | 0.796 | 0.393 | 0.392 | VLM-R1-OVD | 0.045 | 0.375 | 0.660 | 0.511 | 0.111 | 0.337 |
| GPT-5 | 0.225 | 0.528 | 0.288 | 0.926 | 0.366 | 0.267 | LLaVA-Next-Vicuna-7B | 0.160 | 0.399 | 0.277 | 0.869 | 0.218 | 0.241 |
| GPT-4.1 Nano | 0.175 | 0.413 | 0.300 | 0.872 | 0.216 | 0.261 | VLM-R1-Math | 0.035 | 0.330 | 0.533 | 0.506 | 0.130 | 0.269 |
| OpenAI o3 | 0.130 | 0.428 | 0.354 | 0.775 | 0.237 | 0.274 | VLM-R1-REC | 0.080 | 0.360 | 0.238 | 0.755 | 0.236 | 0.180 |
| Claude 3.5 Sonnet | 0.030 | 0.191 | 0.203 | 0.620 | 0.082 | 0.126 | Kimi-VL-A3B-Thk-2506 | 0.010 | 0.203 | 0.439 | 0.388 | 0.057 | 0.170 |
| GPT-5 mini | 0.030 | 0.186 | 0.175 | 0.644 | 0.090 | 0.112 | Deepseek VL2-Small | 0.065 | 0.286 | 0.164 | 0.790 | 0.187 | 0.130 |
| Claude 3.7 Sonnet | 0.035 | 0.136 | 0.144 | 0.702 | 0.041 | 0.101 | Deepseek VL2-Tiny | 0.030 | 0.161 | 0.095 | 0.750 | 0.100 | 0.071 |

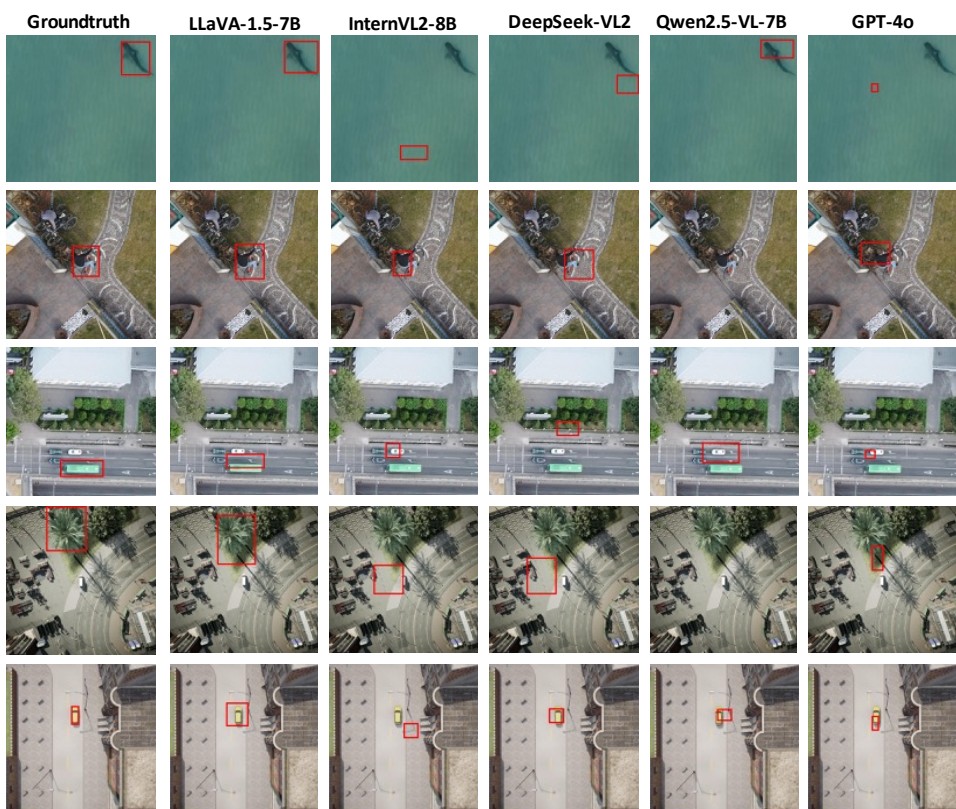

Figure 11: Grounding results from various models.

Table 10: Visual Grounding IoU vs Centroid Containment Comparison.

| Metric | GPT 4o | GPT 4o mini | Gemini 1.5 pro | Gemini 1.5 flash | Claude 3.5 sonnet | Claude 3.5 haiku |
|---|---|---|---|---|---|---|
| Average IoU | 0.05 | 0.03 | **0.40** | 0.25 | 0.07 | 0.06 |
| Centroid Performance (%) | 1.50 | 1.60 | **36.40** | 24.10 | 2.80 | 1.10 |

| Metric | DeepSeek VL2-small | DeepSeek VL2-tiny | LLaVA-Next Qwen-7B | LLaVA-Next Qwen-0.5B | LLaVA 1.5-7B |
|---|---|---|---|---|---|
| Average IoU | 0.09 | 0.08 | 0.06 | 0.05 | **0.35** |
| Centroid Performance (%) | 1.80 | 2.60 | 0.60 | 0.50 | **36.50** |

| Metric | Qwen2.5 VL-7B | Qwen2.5 VL-3B | InternVL2 8B | InternVL2 4B | Phi4 |
|---|---|---|---|---|---|
| Average IoU | 0.07 | 0.02 | 0.04 | 0.02 | 0.01 |
| Centroid Performance (%) | 0.60 | 0.00 | 0.50 | 0.00 | 0.00 |

