# OpenReview forum: "TDBench: Benchmarking Vision Language Models on Top-Down Image Understanding"
_ICLR.cc/2026/Conference — Submitted to ICLR 2026_

### Official Review · Reviewer_TaVy · 2025-10-18

**Soundness:** 2
**Presentation:** 2
**Contribution:** 1
**Rating:** 2
**Confidence:** 4

**Summary:**

The paper introduces TDBench, a new Q&A benchmark for top-down imagery. The dataset is constructed from existing image sources and includes around 2,000 curated questions. The authors also propose a new metric, RotationalEval (RE), designed to evaluate rotation-invariant reasoning by requiring correct answers across four orientations (0°, 90°, 180°, 270°). Using this benchmark, the paper evaluates multiple models and analyzes performance differences through equations relating RE, VE (Vanilla Evaluation), and MA (wrong answers). In addition, the authors conduct four case studies, including digital zoom, physical zoom, occlusion, and z-axis depth understanding, to further explore model behavior under different visual conditions.

**Strengths:**

- The proposed metric effectively leverages the rotation-invariant property of top-down imagery, which is an interesting and meaningful perspective. However, it is somewhat limiting that the metric only considers four discrete rotation angles (0°, 90°, 180°, 270°). Since the paper uses drones as an example, one might expect continuous viewing angles in real-world scenarios, where improving RE may not necessarily correlate with real-world performance.
- The comprehensive performance analysis across various models is impressive

**Weaknesses:**

- As mentioned in the paper, a satellite image can be considered a representative example of top-down images. However, it remains unclear whether the analyses obtained through the proposed TDBench provide new insights compared to existing studies that analyze similar domains with satellite images using vision-language models (VLMs).
- The statistics of the proposed benchmark are not clearly presented. It is difficult to determine how many image–QA pairs exist for each of the ten tasks, and how real versus synthetic data are distributed across them. For example, Sec. B.2 mentions 2,200 images, while Sec. 3.4 refers to 2,000 problems. If space limitations in the main paper prevented including these details, it would be helpful to at least provide a summary table in the appendix.
- The dataset consists of images (I), questions (Q), choices (C), and labels (L). While the images are sourced from existing datasets, the paper lacks a clear explanation of how Q, C, and L were generated.
- In Figure 9 (Sec. B.1), there appear to be four task types (object localization, attribute recognition, visual grounding, spatial relationship) that are not rotation-invariant, as the correct answer changes upon rotation. It is unclear whether new choices and labels were constructed for each rotation case in such situations.
- The analysis in Sec. 4.3 is interesting but seems to rely on strong assumptions when formulating relationships among RE, VE, and MA. The notion of distinguishing questions as known or unknown is ambiguous. Does it refer to whether the image–question–answer pair appeared in training, or something else? Moreover, it is questionable whether rotated images can truly be treated as independent samples, and the meaning of conditionality in this context is unclear. While the equations may reveal some trends, it is debatable whether they support the claim of a “deeper analysis.”
- Although four case studies are presented, only the fourth one appears to be specific to top-down imagery. The others (digital zoom, physical zoom, occlusion) could apply equally to general spatial imagery, and therefore do not provide novel or top-down–specific insights.

**Questions:**

Please refer to Weaknesses.

---

> ### Author Response · Authors · 2025-11-21
>
> We thank the reviewer for the detailed feedback. We address the concerns regarding novelty, dataset construction, and the validity of our probabilistic framework below.
>
> **W1. Novelty vs. Satellite-Image Benchmarks**
>
> Satellite benchmarks typically target low-resolution, meter-level imagery for coarse classification. TDBench targets near-surface, human-scale drone imagery (sub-meter resolution), requiring fine-grained reasoning. To empirically prove this domain gap, we evaluated **GeoChat-7B** (a Remote Sensing VLM) on TDBench against **LLaVA-1.5-7B**.
>
> As shown below, while the GeoChat-7B is perfect at simple detection (Presence: 1.00), it **catastrophically fails** at other reasoning tasks. This confirms that TDBench covers a distinct domain requiring spatial logic that current satellite-based models do not possess.
>
> | **Category** | **LLaVA-1.5-7B RE (VE)** | **GeoChat-7B RE (VE)** |
> | --- | --- | --- |
> | **Object Presence** | 0.32 (0.56) | **1.00 (1.00)** |
> | **Visual Grounding** | **0.37 (0.67)** | 0.00 (0.01) |
> | **Scene Understanding** | **0.81 (0.91)** | 0.14 (0.31) |
> | **Spatial Relationship** | **0.04 (0.41)** | 0.00 (0.26) |
> | **Attribute Comparison** | **0.64 (0.67)** | 0.59 (0.64) |
> | **Attribute Recognition** | **0.39 (0.59)** | 0.14(0.32) |
> | **Dynamic Temporal** | 0.09 **(0.38)** | **0.16** (0.29) |
> | **Hallucination Detection** | 0.28 (0.56) | **0.40 (0.63)** |
> | **Object Counting** | **0.18 (0.36)** | 0.04 (0.26) |
> | **Object Localization** | **0.12 (0.55)** | 0.10 (0.24) |
>
> **W2. Clarification of Dataset Statistics**
>
> We apologize for the confusion. The numbers are consistent but were presented ways. The dataset contains **4,100 questions in total**:
> - **Main Benchmark:** 2,000 questions (200 per category).
> - **Case Studies:** 2,100 questions (used for zoom/altitude/occlusion tests).
> - **Table 3 Confusion:** The confusion comes from the fact that 200 real-world images from the initial pool were allocated to the Case Studies rather than the Main Benchmark. We will add a "Dataset Breakdown" in the Appendix to make this partition explicit.
>
> **W3. Dataset Generation**
>
> We employed a hybrid pipeline:
> - **Real-World Data:** 100% human-annotated via a two-stage custom interface. Stage 1: Crop the original image to perfect square. Stage 2: Annotate Question, Choices, and BBox.
> - **Simulation:** Ground truth (counts, depth, bounding boxes) is extracted programmatically from the engine to ensure zero label noise.
>     - Choices: correct answers are generated by code, while other choices are written by human. We will move these details from the Appendix to the main text when the space permits.
>
> **W4. Handling Rotation-Variant Tasks**
>
> As described in Appendix B.3, we classify questions as "rotation-invariant" or "rotation-sensitive." For sensitive tasks (e.g., Spatial Relationships), the pipeline automatically synchronizes directional references.
>
> • *Example:* If the answer at $0^{\circ}$ is "Top-Left," the system automatically remaps the label to "Top-Right" for the $90^{\circ}$ case.
> New choices/labels are indeed constructed for every rotation; the model is evaluated against each rotation separately without any knowledge of the others.
>
> **W5. Independence Assumption in Reliability Analysis**
>
> The independence assumption is valid because the **model is stateless**.
>
> - **Experimental Isolation:** We evaluate each rotation as a completely separate, zero-shot API call. The model has no memory of the previous orientation.
> - **Justification for $g^4$:** Since the model cannot "remember" the scene, if it does not truly "know" the answer ($\theta=0$), its probability of guessing correctly 4 times in a row is strictly the product of independent probabilities ($g \times g \times g \times g$).
> - **"Known" Definition:** "Known" does *not* mean "seen in training." It is a latent variable representing **Robust Semantic Understanding**. If a model "knows" a concept (e.g., "this is a car"), it should be robust to camera rotations. If it fails on rotation, our framework classifies that not as "robust knowledge", but as a fragile prediction likely derived from overfitting or lucky priors.
>
> **W6. Specificity of Case Studies**
> The four case studies are motivated directly by real-world deployment scenarios for top-down systems such as UAV patrols, rather than by aiming for methodological novelty. In practical applications, understanding how VLMs behave under digital zoom, physical altitude changes, and partial occlusion is critical for determining operational parameters such as cruising height, required resolution, and the conditions under which the system may fail. Our case-study datasets are therefore designed as a **deployment-oriented evaluation suite**, complementing the main benchmark by exposing failure modes that practitioners must consider when using VLMs in top-down settings (e.g., how high a drone can fly before semantic information becomes unrecoverable, or how much occlusion is tolerable during flight).

---

> ### Author Response · Authors · 2025-11-27
>
> Dear Reviewer,
> Thank you for your thoughtful and detailed comments. We have included the clarfications such as statistics, question design, and case studies, as well as some addition experiment result we would like to show to you.  As the discussion period is ending soon, please let us know if there are any remaining issues or points we can further clarify. We truly appreciate your feedback.

---

### Official Review · Reviewer_qai1 · 2025-10-26

**Soundness:** 2
**Presentation:** 3
**Contribution:** 2
**Rating:** 4
**Confidence:** 2

**Summary:**

The paper proposes a new benchmark for top-down image understanding. Current LVLMs are primarily trained and evaluated on frontal views; consequently, their performance on top-down views is worse than on frontal views. To evaluate this gap, the TDB benchmark is introduced, along with additional metrics such as rotation-invariant evaluation and probability-based knowledge reliability analysis.

**Strengths:**

1. Evaluating LVLMs from a top-down view is novel, and there are many critical applications in this domain.

2. The dataset covers a wide range of tasks from a top-down perspective.

3. The motivation is clear, and the paper is easy to follow.

**Weaknesses:**

1. The contribution seems limited, as the paper primarily focuses on evaluating different LVLMs on the proposed TDB benchmark.

2. Providing more insights into why model performance degrades under the top-down view would be beneficial. An initial attempt to address this issue could further strengthen the paper.

3. The paper lacks comparisons with domain-specific models. For example, reporting the performance of state-of-the-art models on selected TDB tasks could help better illustrate the performance gap.

**Questions:**

What is the performance of state-of-the-art models on selected TDB benchmarks?

If domain-specific models perform well on this data, would it be possible to use them to process the images and then feed the results to LVLMs or LLMs for further analysis?

---

> ### Author Response · Authors · 2025-11-21
>
> We thank the reviewer for finding our motivation clear and recognizing the critical applications of this domain. We hope the following new experiments and clarifications regarding domain-specific models will address your concerns.
>
> **W1. Contribution Scope**
>
> While TDBench is a dataset contribution, its primary goal is to establish a rigorous evaluation standard for an underexplored domain. Beyond the data itself, we introduce two methodological contributions:
>
> 1. **RotationalEval (RE):** A metric that enforces physical consistency (invariance), filtering out the "lucky guesses" that inflate standard scores.
> 2. **Reliability Analysis:** A probabilistic framework (Sec 4.3) that mathematically decouples "knowledge" from "guessing," providing a generalizable lens for measuring VLM trustworthiness.
>
> **W2. Insights on Performance Degradation**
>
> The performance drop in top-down views stems from two key factors:
>
> 1. **Data Scarcity:** As noted in the introduction (L42–45), fewer than **7%** of images in VisDrone are truly top-down, and this ratio is even lower in general pre-training data, leading to weak feature alignment for top-down views.
> 2. **Z-axis Compression:** Top-down views compress vertical depth. Our Case Study 4 shows that VLMs fail to distinguish vertical layers (e.g., overpass vs. road) in 2D projections, often hallucinating spatial relationships based on simple pixel proximity rather than 3D understanding.
>
> **W3/Q2. Domain-specific models & Hybrid Pipelines**
>
> We performed two extensive new experiments to address your question about leveraging domain-specific models.
>
> **(1) Domain-Specific VLM (GeoChat)**
> We evaluated **GeoChat-7B** (fine-tuned on remote sensing data) against the general-purpose **LLaVA-1.5-7B**.
> **Result:** As shown below, domain-specific tuning creates a "specialist" that excels at simple detection but catastrophically forgets broader reasoning capabilities.
>
> | **Category** | **LLaVA-1.5-7B (RE)** | **GeoChat-7B (RE)** | **Insight** |
> | --- | --- | --- | --- |
> | **Object Presence** | 0.32 | **1.00** | GeoChat is excellent at detection. |
> | **Spatial Relationship** | **0.04** | 0.00 | GeoChat loses spatial reasoning. |
> | **Attr. Comparison** | **0.64** | 0.59 | Generalist models reason better. |
> | **Dynamic Temporal** | 0.09 | **0.16** | Both struggle significantly. |
> | **Scene Understanding** | **0.81** | 0.14 | GeoChat loses general semantic knowledge. |
> | **Visual Grounding** | **0.37** | 0.00 | GeoChat loses coordinate reasoning. |
>
> **(2) Detection-Assisted Pipeline (YOLO + GPT-4o)**
> To test if a hybrid approach works, we trained **YOLOv11x** on a subset of TDBench data (achieving high robustness, VE=0.857, RE=0.791) and fed images with **overlaid bounding boxes** into GPT-4o. We compare the performance on two categories (Object Localization and Object Presence) and subsets where the detector successfully triggered (`det_count > 0`). Since YOLO11 is not an open-vocab detection model, we filtered the questions that is asking about objects out of trained labels. Thus, there are 161 questions retained (161x4=644samples) in object localization, and 133 questions in object presence.
>
> | Task | Metric | Detection-Augmented | Direct VLM |
> | --- | --- | --- | --- |
> | **Object Localization** *(161 Qs)* | RE | 0.2484 | **0.2919** |
> | *(644 Qs)* | VE | **0.6894** | 0.6770 |
> | *Subset (det_count > 0) (118 Qs)* | acc | 0.6711 | **0.6787** |
> | **Object Presence (133 Qs)** | RE | **0.6466** | 0.6090 |
> | *(532 Qs)* | VE | **0.8346** | 0.7820 |
> | *Subset (det_count > 0) (280 Qs)* | acc | **0.9179** | 0.8286 |
>
> (Note: When `det_count = 0`, no boxes are drawn on the raw images. Two Approaches have identical or very close results.)
>
> **Observations:**
>
> - **Detection helps "Existence" (Presence):** When YOLO detects an object, it provides a massive boost to the VLM’s ability to confirm presence (**91.8% vs 82.9%**). The strong visual cue resolves ambiguity.
> - **Visual aids hinder "Positioning" (Localization):** However, for *Object Localization* (mapping objects to a 3x3 grid), the augmented pipeline performs **worse**.
>     - **Consistency drops:** RE drops significantly (0.25 vs 0.29), indicating the model becomes less robust to rotation.
>     - **Accuracy drops:** Even when the object is found and boxed (`det_count > 0`), the VLM is *less* accurate at placing it in the correct grid cell (0.671) compared to the raw image (0.679).
>
> **Q1. Performance of SOTA Models**
>
> As detailed in Table 1 and Figure 3b, state-of-the-art models (GPT-5, Gemini 2.5 Pro) achieve high Vanilla accuracy ($VE > 75\%$) but suffer substantial drops under RotationalEval ($RE \approx 61.1\%$). Even the best model (Gemini 2.5 Pro) lags significantly behind the **Human Baseline (VE 0.92 / RE 0.89)** established in our new human study on randomly sampled 100 questions on average.

---

> ### Author Response · Authors · 2025-11-27
>
> Dear Reviewer,
> Thank you again for your detailed feedback. We have added clarifications and supplementary analyses addressing the points you raised. As the discussion period is ending soon, please let us know if there are any remaining questions or aspects you would like us to elaborate on. Your input has greatly improved the clarity and completeness of the paper.

---

### Official Review · Reviewer_xDnU · 2025-10-31

**Soundness:** 3
**Presentation:** 3
**Contribution:** 3
**Rating:** 4
**Confidence:** 4

**Summary:**

This paper presents TDBench, a new benchmark designed to evaluate Vision–Language Models (VLMs) in top-down image understanding scenarios, such as aerial and autonomous navigation views. The benchmark includes 2,000 curated QA pairs per rotation angle, and introduces RotationalEval (RE), an evaluation metric that measures answer consistency across four rotated views of the same scene. The authors further propose a reliability analysis framework to separate genuine understanding from chance-level responses. Extensive experiments across major VLMs (GPT-4V, Gemini, Claude, InternVL, Qwen-VL, etc.) show that existing models perform poorly in rotational robustness and spatial consistency. Overall, the paper highlights an underexplored but practically important perspective in multimodal evaluation: understanding and reliability in top-down perception.

**Strengths:**

1. The benchmark addresses an overlooked setting (top-down perception), which is practically relevant for safety-critical applications such as aerial surveillance and navigation.
2. The dataset and evaluation framework are clearly described and carefully curated, with attention to rotation invariance and visual grounding reliability.
3. The authors evaluate a broad range of state-of-the-art VLMs and provide detailed error and reliability analyses.
4. The paper is well organized, with strong motivation, readable methodology, and informative figures.

**Weaknesses:**

1. The core contributions are primarily in dataset and evaluation design rather than algorithmic or theoretical advances. RotationalEval is intuitive and could be viewed as a straightforward extension of consistency testing.
2. The reliability framework is described conceptually but lacks a formal quantitative or probabilistic definition of “genuine knowledge vs. chance.”
3. While rotation is important, other forms of geometric variation (scale, translation, occlusion) are not addressed. The study could be more comprehensive in analyzing general spatial invariance.
4. Some question templates and image sources are limited in semantic diversity, which might restrict the generalizability of conclusions.

**Questions:**

1. How does RotationalEval handle questions that are not rotation-invariant (e.g., “What is on the left/right side of the image?”)?
2. Can the proposed reliability framework be formalized mathematically, rather than qualitatively distinguishing “genuine” vs. “lucky” answers?
3. Have the authors considered evaluating compositional reasoning or anomaly reasoning tasks in aerial views to complement spatial consistency?
4. How well do current reasoning-based models perform on this task? Could the paper include a more detailed comparison or discussion of reasoning-centric approaches and their limitations in contextual anomaly understanding within top-down scenes?
5. How scalable is TDBench？Can it be extended to larger multimodal corpora or fine-tuned models?

---

> ### Author Response · Authors · 2025-11-21
>
> We thank the reviewer for recognizing the practical relevance of TDBench for safety-critical applications and for highlighting the clarity of our evaluation framework. We address your concerns below.
>
> **W1. Contribution beyond evaluation**
>
> While TDBench is primarily a benchmark contribution, we argue that **RotationalEval (RE)** and our **Reliability Analysis** are methodological contributions that help measure VLM trustworthiness. Rather than relying on static accuracy (which we show is inflated by up to ~20% due to guessing), our framework provides a diagnostic probe into model consistency and "knowledge", offering a new approach for evaluating reliability that extends beyond this specific dataset.
>
> **W2 & Q2. Mathematical Formalization of Reliability**
>
> We respectfully point out that it is formalized mathematically in **Section 4.3** and **Appendix C.**
>
> - **System of Equations:** We model reliability as a probabilistic mixture where observable statistics ($RE, VE, MA$) are non-linearly related to latent parameters ($\theta, r, g$) via the system derived in Eq. 1.
> - **Closed-Form Solution:** In **Appendix C**, we prove that these parameters are generically unique by reducing the system to a cubic polynomial and analyzing the discriminant. This provides a rigorous, quantitative solution for "genuine knowledge" ($\theta$), rather than a qualitative heuristic. We will make the pointer to Appendix C more prominent in the main text.
>
> **W3. Geometric Variations (Rotation vs. Scale/Flip)**
> We focused on rotation because it is a transformation that changes the pixel representation while **preserving the exact semantic content** of the scene.
>
> - **Mirror Flips:** These often violate physical ground truth in aerial scenes (e.g., traffic flow direction), making the ground truth ambiguous.
> - **Scale/Occlusion:** We do analyze these in **Case Studies 1 & 3** (Sec 5.1, 5.3). However, we distinguish them from rotation: zooming or cropping inherently changes the *information content* (objects leave the frame), testing **robustness**, whereas rotation tests **invariance** (the information remains identical).
>
> **W4 & Q1. Handling Rotation-Variant Questions**
>
> As detailed in **Appendix B.3**, we employ **Rotation-Aware Question Design**. We use placeholder tokens in the ground truth that automatically synchronize directional references (e.g., mapping "top-left" $\to$ "top-right" after a 90° rotation). This ensures the ground truth remains physically correct across all rotations. Unlike many existing benchmarks that rely on rigid templates with simple object substitution (e.g., POPE), TDBench incorporates multiple task types and varied linguistic structures to assess diverse spatial and reasoning contexts."
>
> **Q3. Compositional and Anomaly Reasoning**
> We agree these are important, and TDBench actually includes them. For example:
>
> - **Compositional:** The "Spatial Relationship" category  requires aggregating relative positions of multiple objects (e.g., "How is the dog positioned relative to the bike?").
> - **Contextual Anomaly:** The "Attribute Comparison" category  tests for physical consistency (e.g., "Are all cars moving in the same direction?"), effectively acting as an anomaly detection task for VLM hallucinations.
>
> **Q4. Performance of Reasoning-Based Models**
>
> We performed a specific analysis grouping the 60 evaluated models into "Reasoning/Thinking" (e.g., o3, InternVL-Thinking) vs. "Standard" models based on our results. Reasoning models improve general accuracy ($VE$) but **do not** solve the rotational robustness gap. They suffer the exact same relative drop in performance.
>
> | **Model Group** | **Avg VE** | **Avg RE** | **Drop %** |
> | --- | --- | --- | --- |
> | **Reasoning** | 0.643 | 0.441 | **-31.4%** |
> | **Standard** | 0.578 | 0.397 | **-31.3%** |
>
> **Q5. Scalability and extensibility**
>
> The benchmark is designed to be model-agnostic and scalable. It is integrated into an open-source evaluation framework, allowing any VLM compatible with standard APIs or HuggingFace interfaces to be evaluated automatically. The benchmark can be extended easily with the pipeline for handling the various rotations.

---

> ### Author Response · Authors · 2025-11-27
>
> Dear Reviewer,
>
> I hope you are doing well. As we approach the end of the discussion period, I wanted to check whether there are any remaining questions or points we can help clarify. We hope our responses have addressed your concerns, and we would be happy to elaborate further if needed. Thank you again for your thoughtful feedbacks.

---

### Official Review · Reviewer_UP7S · 2025-11-01

**Soundness:** 3
**Presentation:** 3
**Contribution:** 3
**Rating:** 8
**Confidence:** 3

**Summary:**

Motivated by a lack of benchmarks dedicated to top-down image understanding for outdoor images, the authors propose TDBench. TDBench is a benchmark of 2000 questions with images drawn from public datasets of drone images and simulated environments. It consists of two question types, one involving multiple-choice questions and another involving bounding box prediction. The dataset features 10 question categories with 200 questions each. A key design feature is "RotationalEval," which evaluates model responses to questions for 4 separate rotations of the image and treats a question as known if answers to all rotations are correct. The authors evaluate a total of 60 vision-language models on the dataset and conduct additional case studies using a held-out dataset of 2100 questions.

**Strengths:**

- The motivation behind the dataset is valid, as there is a lack of benchmarks for top-down image understanding that do not place object detection as their focus.
- The RotationEval setting is a clever way to mitigate the impact of random guessing on a dataset that relies on multiple choice questions for evaluation.
- The authors evaluate a broad range of models, both proprietary and open, of various sizes (60 total). Figure 3 is acts as a very clear way of demonstrating results and scaling trends, particularly when there are this many models.
- The data dedicated to case studies help assess the impact of various features such as drone altitude or the proportion of the target object in the image.

**Weaknesses:**

- While the authors note in the appendix that the dataset was filtered for correctness, there are two analyses that I think would help verify the quality of the dataset. Firstly, there should be a text-only evaluation without any images to ensure that the dataset (or particular question categories) are not solvable using guesswork or spurious correlations. Secondly, human accuracy should be provided, at the very least for a subset of questions, to have an oracle estimate of accuracy.
- It was somewhat difficult to parse the definitions in Section 4.3. Particularly, I'm not sure what the condition is for a question to be "known" or "guessed." In the framing of the RotationEval section, guessing implies inconsistent answers for the rotations. If so, how do "known" questions not have an accuracy of 100%?
- The key contribution of TDBench is providing a benchmark for top-down image understanding that does not rely purely object detection. At the same time, however, questions categories such as Object Presence, Counting, Localization, Visual Grounding and Spatial Relationship effectively rely on the exact same set of skills as object detection but render them in QA form.

**Questions:**

- How was the preliminary audit for VisDrone conducted?

---

> ### Author Response · Authors · 2025-11-21
>
> We thank the reviewer for the encouraging assessment, and for recognizing the value of TDBench’s RotationEval design and scaling analysis. We have addressed the suggestions for verification and clarification below.
>
> **W1. Dataset verification (Text-only baseline and Human accuracy)**
>
> We performed two additional experiments on 9 tasks in TDBench (excluding visual grounding):
>
> - **Text-Only Baseline:** We evaluated three models (GPT-4.1, GPT-5, Qwen2.5-VL-7B) without providing images. These models achieved **33.2% – 35.4% on VanillaEval (VE) and 17.2% – 17.8% on RotationalEval (RE).** Given the mix of 2-choice and 4-choice questions, the random-guess baseline is **$\approx 30.6\%$**. This near-random performance confirms that the questions cannot be solved via language priors or spurious text correlations alone.
> - **Human Study:** We randomly sampled 100 questions evaluated under all four rotations (400 instances). Human subjects achieved **0.92 VE** and **0.89 RE on average**. This indicates:
>     - *(i)* **Solvability***:* The high accuracy confirms the questions are unambiguous (errors were primarily from difficult object counting problems).
>     - *(ii)* **Validity of RE:** The minimal gap between human VE and RE (0.03) contrasts sharply with the large gaps seen in VLMs, validating RE as a strong metric for assessing model trustworthiness vs. hallucination.
>
> We will add these results to Table 1 or in text in the revision.
>
> **W2. Clarification of “known” vs. “guessed” (Section 4.3)**
>
> We appreciate the opportunity to clarify the probabilistic framework (Eq. 1). In our model, "knowledge" ($\theta$) is a latent state, while the answer is an observed outcome. Even when a model "knows" a concept, it may not answer correctly 100% of the time ($r < 1$). While our experiments used greedy decoding, inputs from 4 rotations are not entirely identical across rotations (1) the image pixel grid changes (rotation) (2) prompt text changes (e.g., "top-left" becomes "top-right").  $r$ captures the robustness of the model's knowledge. We will clarify in Section 4.3 that "known" implies a high probability of correctness ($r$) across varying valid views, but not necessarily a deterministic 1.0.
>
> **W3. Relation to Object Detection**
>
> While several categories involve object recognition, TDBench evaluates skills distinct from pure detection. To quantify this, we computed Pearson correlations across 60 models among the five categories you mentioned. The correlations are generally weak (average $\rho = 0.38$ for VE). Localization and Spatial Relationship show higher coupling ($\rho = 0.83$), however they are not identical. For example:
>
> - **LLaVA-1.5-13B** scores high on Localization (0.755) but poor on Spatial reasoning (0.485)
> - **Kimi-VL-Thinking** shows the inverse (Spatial=0.63, Loc=0.43).
>
> This indicate that although they show high coupling, they can measure different abilities of VLMs.
>
> **Q1. Preliminary audit for VisDrone**
> As noted in the introduction, fewer than 7% of VisDrone images were suitable. The audit was conducted by manually screening the dataset image-by-image. We discarded oblique, side-view, and horizon-level shots, retaining only those with clear downward-looking (top-down) perspectives to ensure domain consistency.

---

> ### Author Response · Authors · 2025-11-27
>
> Dear Reviewer,
>
> Thank you again for your thoughtful and encouraging review. As the discussion period is closing, please let us know if there is anything further we can clarify regarding the additions we made. We appreciate your time and are happy to provide any additional details if needed.

---

### Meta-Review · Area_Chair_4R7w · 2026-01-08

**Summary:**

This paper receives 4 high-quality reviews, with 1 positive initial ratings (8) and 3 negative initial rating (4, 4, 2).

No reviewer has participated in the rebuttal discussions.

The 3 negative reviewers have many concerns regarding to: limited contribution (pointed out by 2 negative reviewers), limited insights and analysis regarding to the specifically studied top-down images (pointed out by 2 negative reviewers), limited comparisons with domain-specific models, limited in semantic diversity, benchmark/task settings, etc.

Overall, the reviewers have too many concerns, and the majority of the reviewers have negative initial impressions of the paper. Though some of the them are addressed through the authors' responses, many concerns are still not fully addressed.

**Reviewer Concerns:**

See the above summary.

**Reviewer Scores:**

See the above summary.

---

### Decision · Program_Chairs · 2026-01-26

Reject